# Beyond Hard and Soft: Hybrid Context Compression for Balancing Local and Global Information Retention

## Abstract

Large Language Models (LLMs) encounter significant challenges in long-sequence inference due to computational inefficiency and redundant processing, driving interest in context compression techniques. Existing methods often rely on token importance to perform hard local compression or encode context into latent representations for soft global compression. However, the former struggles to retain global information, while the latter struggles to maintain local details. To address this, we propose **Hy**brid **Co**ntext **Co**mpression (HyCo$_2$) for LLMs, which integrates both global and local perspectives to guide context compression while retaining both the essential semantics and critical details for task completion. Specifically, we employ a hybrid adapter to refine global semantics with the global view, based on the observation that different adapters excel at different tasks. Then we incorporate a classification layer that assigns a retention probability to each context token based on the local view, determining whether it should be retained or discarded. To foster a balanced integration of global and local compression, we introduce auxiliary paraphrasing and completion pretraining before instruction tuning. This promotes a synergistic integration that emphasizes instruction-relevant information while preserving essential local details, ultimately balancing local and global information retention in context compression. Experiments show that our HyCo$_2$ method significantly enhances long-text reasoning while reducing token usage. It improves the performance of various LLM series by an average of 13.1% across seven knowledge-intensive QA benchmarks and LongBench. Moreover, HyCo$_2$ matches the performance of uncompressed methods while reducing token consumption by 88.8%. Our code will be available at https://anonymous.4open.science/r/HyCo2.

## 1 Introduction

Large Language Models (LLMs) (Achiam et al., 2023; Dubey et al., 2024) exhibit strong performance across a wide range of tasks. Effective handling of extended context is crucial for applications such as retrieval-augmented generation (RAG) (Gutiérrez et al., 2024; Zhao et al., 2024), long-term memory systems (Hu et al., 2024; Wang et al., 2024), and complex reasoning frameworks (Liao et al., 2025b; Lightman et al., 2023). However, for long-form textual inputs (Liu et al., 2025a; Xu et al., 2025; Zhang et al., 2024a), LLMs still face fundamental bottlenecks: 1) **Computational Inefficiency**: The quadratic complexity of self-attention (Xia et al., 2024) results in $\mathcal{O}(N^2)$ FLOPs, which significantly increases latency and cost for inputs with tens of thousands of tokens (Liao et al., 2025c). 2) **Information Dilution**: Noisy or redundant context weakens the model's ability to focus on task-relevant content (Liu et al., 2024; Shi et al., 2023; Wu et al., 2024), especially in RAG where retrieved documents often contain irrelevant segments. 3) **Context Window Limitations**: The fixed input bounds of current models (e.g., 8k/32k tokens for most 7B LLMs) hinder their ability to process extended inputs such as full research papers or multi-document corpora.

Context compression has emerged as a promising solution to the challenges of long-context processing by selectively preserving critical information while reducing computational overhead (Chang et al., 2024). Existing methods mainly fall into *hard* and *soft* compression, each involving inherent trade-offs among efficiency, detail, and semantic preservation (Figure 1 (a)). Hard compression selects

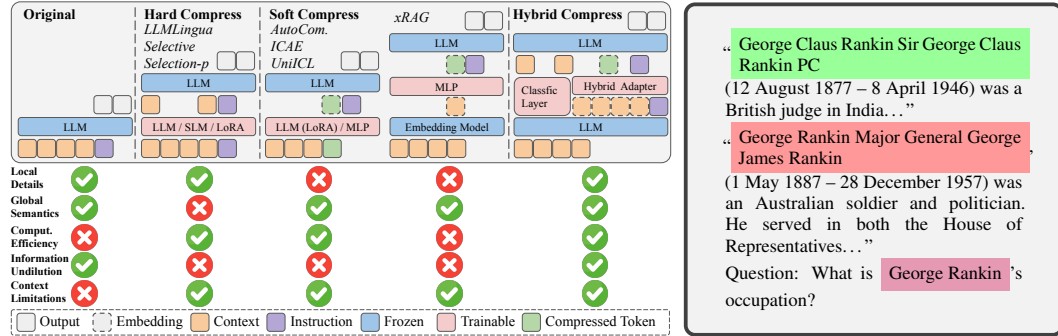

Figure 1: (a) **Comparison of Different paradigms for processing long context**: original input, hard compression, soft compression and our hybrid compression (b) An illustration of the *George Rankin* occupation disambiguation task. Further analysis is provided in Appendix A.

natural language segments based on metrics like logits or perplexity (Li et al., 2023b), but often sacrifices fluency, coherence, and the handling of removed context (Shandilya et al., 2024), while its reliance on chunking increases time complexity (Chuang et al., 2024; Jiang et al., 2024). Conversely, soft compression encodes text into dense latent representations for higher compression rates and scalability (Chevalier et al., 2023). However, this approach disrupts sequential structure, neglects local details, reduces interpretability, and complicates information tracing (Deng et al., 2024).

However, striking the right balance between information preservation and efficiency remains difficult. As shown in Figure 1 (b), the *George Rankin* disambiguation task paired with a document describing two individuals: (1) Sir George Claus Rankin (British judge) and (2) Major General George James Rankin (Australian soldier/politician). Hard compression methods (e.g., LLMLingua2 (Pan et al., 2024)) often discard one entity entirely, while soft compression approaches (e.g., xRAG (Cheng et al., 2024)) reduce content to a few latent tokens, omitting crucial details such as titles or nationalities. Both approaches fail to support correct reasoning. Therefore, an effective solution requires achieving a balance across several critical dimensions: 1) **Local Detail Preservation**, accurately retaining salient information units without introducing redundancy; 2) **Global Semantic Completeness**, capturing the core meaning of the document while maintaining contextual coherence and avoiding semantic loss. Given these limitations, a key research question arises: *Can we combine the specificity of explicit tokens with the abstraction of latent representations to achieve a balance between local detail and global information retention in context compression?*

To overcome the limitations above, we introduce $HyCo_2$, a hybrid context compression method that jointly preserves global semantics and local details for long-context reasoning. Our design is motivated by how humans process information: starting from a coarse global understanding and progressively refining into fine-grained details (Figure 1 (a), right). **Global Compression** leverages a *hybrid adapter*, combining the strengths of MLPs (Liu et al., 2023), Q-Former (Li et al., 2023a), and Resampler (Alayrac et al., 2022), which captures overarching contextual information through joint local and global attention mechanisms: Local MLP segments the input context into groups and compresses each group into a single token, maintaining structural coherence and emphasizing subregions. Global QFormer utilizes learnable tokens that interact with both the instruction and the entire context to extract key global semantics. **Local Compression** employs an auxiliary classification layer trained to identify and retain critical tokens (Chung et al., 2024), ensuring fine-grained details necessary for accurate reasoning are preserved. The outputs of local and global attention are then softly fused, producing a rich, instruction-aware representation that is subsequently passed to the frozen LLM. However, we find that training the global and local compression modules simultaneously presents a significant challenge. To overcome this and fully leverage $HyCo_2$'s potential, we propose pretraining global and local compression modules using paraphrase and completion tasks, respectively, before instruction tuning. This alternating training strategy enables effective learning and utilization of both global and local representations. Extensive empirical studies validate the effectiveness of our $HyCo_2$. Remarkably, our approach achieves leading performance across various models on 7 QA datasets and LongBench (Bai et al., 2024) with significantly fewer costs, even matching the performance of the original context. Our key contributions are summarized as follows:

- We propose HyCo$_2$, a hybrid context compression method for LLMs that unifies hard (local token selection) and soft (global latent encoding) compression. HyCo$_2$ effectively reduces computational costs while enabling efficient understanding of long context.

- HyCo$_2$ employs minimal parameter updates and ensures lightweight training and inference by avoiding reliance on external, pre-trained models for compression.

- We propose an alternating pretraining strategy that decouples global abstraction and local retention before joint alignment, improving stability and sample efficiency.

- Extensive experiments on multiple benchmarks show that HyCo$_2$ achieves superior performance compared to existing methods with significantly lower computational overhead, thereby offering valuable insights into designing effective hybrid context compression strategies for LLMs.

## 2 METHODOLOGY

This section first reviews foundational concepts in LLM context compression (Sec. 2.1) and why we need to ues the soft mixture-of-experts (MoE). We then detail our approach, HyCo$_2$ (Sec. 2.3), which uses the MOE mechanism for global context refinement and a classification layer for local hard compression. Next, Section 2.4 introduces an alternating training strategy to align the compressed text with the LLM's semantic space. Figure 3 illustrates the model architecture and training workflow.

### 2.1 PRELIMINARIES

Context compression aims to reduce the length of input while preserving its functional utility in guiding LLMs to perform downstream tasks effectively. This is particularly important as the complexity of tasks increases, necessitating longer context that can lead to higher memory usage and slower inference. Formally, given a context represented as a sequence of tokens $\boldsymbol{x} = (x_1, x_2, \ldots, x_N)$, where $N = |\boldsymbol{x}|$ denotes the sequence length, the objective of context compression is to identify a shorter sequence $\hat{\boldsymbol{x}}$ such that:

$$\min_{\hat{\boldsymbol{x}}} \mathcal{D}(f(\cdot|\boldsymbol{x}), f(\cdot|\hat{\boldsymbol{x}})), \quad \text{s.t. } |\hat{\boldsymbol{x}}| \leq |\boldsymbol{x}| \tag{1}$$

where $f(\cdot|\boldsymbol{x})$ represents the conditional distribution over the original context $\boldsymbol{x}$, $f(\cdot|\hat{\boldsymbol{x}})$ represents the conditional distribution over the compressed context $\hat{\boldsymbol{x}}$, and $\mathcal{D}$ is a divergence metric (e.g., Kullback-Leibler divergence) that quantifies the difference between the two distributions. The goal is to minimize $\mathcal{D}$, ensuring that the compressed $\hat{\boldsymbol{x}}$ retains essential information from the original $\boldsymbol{x}$.

### 2.2 WHY SOFT MIXTURE OF EXPERTS?

Current paradigms for long-context compression face a fundamental dilemma, presenting a trade-off between global semantics and local details. To bridge this gap, we draw inspiration from human information processing mechanisms: when humans analyze long texts, they first form a coarse-grained global understanding (e.g., 'this document compares two George Rankins') and then zoom in on fine-grained details (e.g., their titles and nationalities). This *global-to-local* cognitive logic motivates HyCo$_2$'s dual-level framework designed to synergistically retain both global semantics and local details rather than trade one for the other.

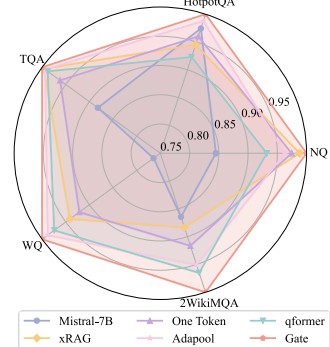

Figure 2: Significance of Soft MoE. The reported values represent the performance ratio of baselines to the best **Gate**.

Our architectural choices are empirically grounded in an ablation study, the results of which are consistent with findings in prior multimodal research (Zhang et al., 2024b). We identify a critical trade-off between the expressive capacity of attention-based modules like QFormer and the efficiency of simpler architectures like MLPs. While QFormer offers greater theoretical flexibility, it often requires extensive hyperparameter tuning and, under a fixed query token budget, can lead to performance degradation. As illustrated in Figure 2, using identical instruction-tuning datasets and training configurations, we systematically compare module designs under fixed query-token budgets. Replacing an MLP-based compressor (i.e., Adapool) with QFormer results in a discernible performance drop across a majority of tasks. This suggests

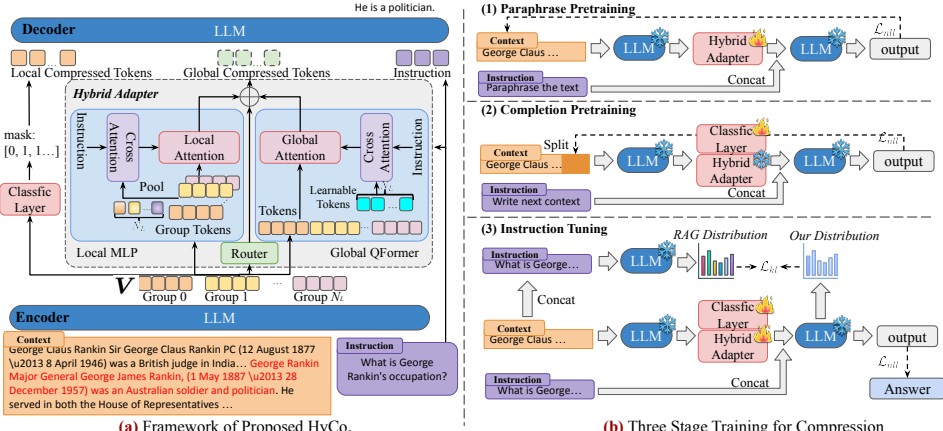

(a) Framework of Proposed HyCo₂      (b) Three Stage Training for Compression

Figure 3: (a) **Hybrid Context Compression Framework.** We employ a classification layer for local token selection and use a hybrid adapter to extract instruction-relevant representation. Additionally, a router optimizes the global context through soft integration, thereby optimizing overall context representation. (b) **Alternating Training Method.** (1) Refining the hybrid adapter with paraphrase pretraining, (2) optimizing the classification layer with completion pretraining and (3) instruction tuning for both the hybrid adapter and the classification layer. We employ a single target LLM that operates in two capacities: as an encoder to compress the original context, and as a decoder that conditions on the encoder's output to execute downstream tasks.

that simpler projection architectures may produce contextual representations that are more readily assimilated by the base LLM. However, on complex tasks requiring fine-grained cross-document analysis, such as the 2WIKI reasoning benchmark, QFormer's dynamic attention mechanism confers a distinct advantage by enabling task-relevant feature selection.

Further investigation reveals that even a single learnable query token (i.e., One Token) can match the performance of sophisticated baselines like xRAG (Cheng et al., 2024). This finding critically underscores the severe information bottleneck of static, MLP-based single-token projections in reasoning-intensive scenarios. Ultimately, the results demonstrate that a Mixture-of-Experts (MoE) framework that gates between QFormer and MLP components achieves robustly superior performance across all tasks, thereby validating our hybrid architectural hypothesis.

## 2.3 HYBRID CONTEXT COMPRESSION

We propose a novel hybrid framework for long-context compression that reconciles global semantic preservation with local detail retention. Our approach integrates a **Hybrid Adapter** (left part of the Fig. 3 (a)) and **Selective Mining Layer** (right part of the Fig. 3. This dual strategy enables efficient context reduction while maintaining both high-level coherence and fine-grained accuracy.

**Soft Global Context Refinement within Hybrid Adapter.** To address the *global semantic loss* of hard compression and *structural incoherence* of soft compression, we propose a novel **hybrid adapter** that optimizes global semantics by synergistically leveraging the strengths of MLPs and QFormer via a **joint local-global attention mechanism**. Specifically, we employ a **noisy mixture-of-experts** (MoE) framework to dynamically adjust the weight of local and global attention based on context and instruction (Zhang et al., 2024b). In this framework, for feature $V \in \mathbb{R}^{S \times D}$ derived from Encoder (i.e., final hidden states of Encoder), where $S$ denotes the input length and $D$ the embedding dimension. To align the inputs for the local and global pathways, we first segment the input features $V$ into $N_L$ contiguous groups $\{V^i\}_{i=1}^{N_L}$, where $N_L$ is the number of learnable queries in the Global QFormer. Each group $V^i$ (where $0 \leq i < N_L$) contains $\lceil S/N_L \rceil$ tokens. We then apply average pooling to each group to obtain a set of pooled representations $V_p = \{V_p^i\}_{i=1}^{N_L}$. These pooled tokens serve as condensed representations for both the gating network and the local attention mechanism. Given a learned gating network $\mathcal{G}$ dynamically determines the fusion weights for the two adapters: $\mathcal{G}(V_p)_0 \cdot f_m(V) + \mathcal{G}(V_p)_1 \cdot f_q(V)$, where $f_m(\cdot)$ and $f_q(\cdot)$ denote the MLPs and QFormer branches,

respectively. We inject **learnable noise** during training to mitigate the gating network's tendency to favor a single adapter disproportionately. This is formalized with a standard normal distribution $\mathcal{N}(0, 1)$, router weight matrix $\mathbf{W}_g$ and noise weight matrix $\mathbf{W}_{\text{noise}}$:

$$\mathcal{G}(\boldsymbol{V}_p) = \text{Softmax}\left(\left\{ (\boldsymbol{V}_p \cdot \mathbf{W}_g)_i + \mathcal{N}(0, 1) \cdot \text{Softplus}\,(\boldsymbol{V}_p \cdot \mathbf{W}_{\text{noise}})_i \right\}_{i=1}^2\right). \tag{2}$$

To enhance instruction awareness, we integrate cross-attention mechanisms with instruction embedding $\boldsymbol{C}$ into both the MLP ($f_m(\cdot)$) and QFormer ($f_q(\cdot)$) branches. For **local attention in the MLP branch**, Then the local attention refines it within each group, ensuring that subregions (e.g., a paragraph about 'Sir George Rankin') remain semantically coherent:

$$f_m(\boldsymbol{V}) = \bigoplus_{i=0}^{\boldsymbol{n}-1} \text{MLP}(\text{Attn}(\underbrace{\text{CrossAttn}(\boldsymbol{V_p^i}, \boldsymbol{C})}_{\text{Query}}, \underbrace{\boldsymbol{V^i}}_{\text{Key}}, \underbrace{\boldsymbol{V^i}}_{\text{Value}})), \tag{3}$$

where $\text{Attn}(\cdot)$ denotes the standard attention mechanism, parameterized by query, key, and value matrices, while $\text{CrossAttn}(\cdot)$ denotes instruction-context fusion. While local attention mechanisms preserve textual structure by restricting focus to localized sub-regions, this approach risks incorporating instructionally irrelevant content within partitioned regions. To mitigate this limitation, we employ the QFormer for **global attention** to dynamically identify and emphasize portions of the context most critical to the given instruction. Specifically, we introduce learnable query tokens $\boldsymbol{L} \in \mathbb{R}^{N_L \times D}$, where $N_L$ denotes the token count. This token set interacts with the instruction embedding $\boldsymbol{C}$ through cross-attention, augmented by standard absolute positional embeddings $\text{Pos}(\cdot)$ positional disambiguation and preserving sequential oreder. Then it attends to the full context $\mathbf{V}$ to capture cross-group relationships, e.g., identifying that the query requires distinguishing between two 'George Rankins' and prioritizing occupation-related details. The global attention is as follows:

$$f_q(\boldsymbol{V}) = \text{Attn}\left(\text{CrossAttn}(\boldsymbol{L}, \boldsymbol{C}), \mathbf{V} + \text{Pos}(\mathbf{V}), \mathbf{V}\right). \tag{4}$$

**Hard Selective Local Context Mining through Classification Layer.** The information content of each token $x_i$ is quantified by a retention probability $\boldsymbol{p}_i \in [0, 1]$, with higher values indicating greater significance. Consistent with previous research (Chung et al., 2024), we avoid designing a separate deep network for this estimation. Instead, we leverage the feature $\boldsymbol{V} = \{\boldsymbol{v}_1, \boldsymbol{v}_2, \ldots, \boldsymbol{v}_n\}$, where $\boldsymbol{v}_i$ corresponds to the token $x_i$. A linear projection layer processes these feature to compute the vector of retention probabilities $\boldsymbol{p} = [\boldsymbol{p}_1, \ldots, \boldsymbol{p}_n]$ via $\boldsymbol{p} = \sigma(\mathbf{W}\boldsymbol{V} + b)$, where $\sigma$ represents the Sigmoid function, ensuring outputs lie within $[0, 1]$. $\mathbf{W}$ and $b$ are the linear layer's weight matrix and bias vector, respectively, which are learned parameters mapping features to probabilities. Based on a target compression ratio (e.g., keeping the Top-$k\%$), tokens associated with the highest $\boldsymbol{p}_i$ values are retained. Furthermore, the generation of $\boldsymbol{p}$ can be integrated into a single forward pass shared with the previously described global compression strategy, thereby reducing computational overhead.

## 2.4 ALTERNATING TRAINING STRATEGY

We designed a three-stage training strategy for the classification layer and hybrid adapter (Figure 3 (b)), motivated by challenges in achieving optimal convergence when training both simultaneously (akin to a bilinear problem (Zhang et al., 2024b)). Stage 1: The hybrid adapter is pre-trained via a paraphrase task to reconstruct context using $\mathcal{G}(\boldsymbol{V})$ by minimizing the negative log-likelihood loss $\mathcal{L}_{\text{nll}}$. Stage 2: With the hybrid adapter frozen, the local compression classification layer undergoes further pre-training using a completion task, also optimizing $\mathcal{L}_{\text{nll}}$. Stage 3: Global and local compression are jointly fine-tuned with Self-Distillation, balancing interactions to better preserve information. This involves minimizing both language modeling loss $\mathcal{L}_{\text{nll}}$ and a KL divergence term $\mathcal{L}_{\text{kl}}$ (Equation 1) against a teacher-student paradigm on a hybrid open-source dataset. Drawing on self-distillation and imitation learning, the teacher, a RAG model with access to the full document, provides target distributions for the student, our HyCo$_2$, which uses only the compressed context. Minimizing this KL divergence compels HyCo$_2$ to replicate the teacher's performance, effectively training it to maximize the information extracted from its compressed representation. The final loss is the linear combination controlled by a hyperparameter: $\mathcal{L}_{\text{nll}} + \alpha\mathcal{L}_{\text{kl}}$. Empirically, single-stage joint training led to suboptimal performance, likely due to the model favoring globally informative but easier features. Our staged approach enforces a structured progression from global representation learning to local compression. Detailed training objectives are provided in Appendix B.

## 3 EXPERIMENTS

### 3.1 EXPERIMENTAL SETUP

**Datasets.** We follow xRAG (Cheng et al., 2024), utilizing 17 datasets for instruction tuning. The retrieval corpus is based on the December 2021 Wikipedia dump, with Contriever (Izacard et al., 2021) as the default retriever. By default, the instruction tuning stage uses the *top-5* retrieved documents, while the downstream evaluation phase uses the *top-3*. For completion pertaining (Stage 2), we use the "2023-06" snapshot from RedPajama-Data-V2 (Weber et al., 2024). We evaluate our method on 7 QA datasets, including 5 open-domain QA datasets: **NaturalQuestions (NQ)** (Kwiatkowski et al., 2019), **TriviaQA (TQA)** (Joshi et al., 2017), **WebQuestions (WQ)** (Berant et al., 2013), **PopQA (PQA)** (Mallen et al., 2023), and **ComplexWebQuestions (CWQ)** (Talmor & Berant, 2018) as well as 2 multi-hop QA datasets: **HotpotQA (HQA)** (Yang et al., 2018) and **2WikiMultihopQA (2WIKI)** (Ho et al., 2020). Following prior work, we use the **Exact Match (EM)** metric to assess performance. For long-context understanding, we employ **LongBench** (Bai et al., 2024) to thoroughly assess HyCo$_2$'s achievable performance.

**Implementation Details.** Evaluations of HyCo$_2$ are conducted using **LLaMA3.1-8B-Instruct** (Dubey et al., 2024), **Qwen2.5-7B-Instruct** (Yang et al., 2024), and **Mistral-7B-Instruct-v0.2** (Jiang et al., 2023a), with the base LLM kept frozen during training. The hybrid adapter and classification layer are randomly initialized. We set the number of query tokens ($N_L$) to 16 and the keeping ratio ($k\%$) to 10% by default. We use the learning rate of 1e-4 at the pretraining stage and 2e-5 in the instruction tuning stage. We train 1 epoch for all stages on 8×NVIDIA A100 GPUs (80GB).

**Baselines.** Since the LLM in our method remains frozen, the selected baselines must support plug-and-play functionality without requiring any alteration to the LLM's parameters (Mu et al., 2023; Wang et al., 2023). Accordingly, we focus on three categories of baselines: 1) Uncompressed : Vanilla, RAG 2) Hard Compression : TF-IDF, LongLLMLingua (Jiang et al., 2024), LLMLingua2 (Pan et al., 2024), EXIT (Hwang et al., 2024) 3) Soft Compression : xRAG (Cheng et al., 2024) More experimental details are in the Appendix C.2.

### 3.2 MAIN RESULTS

**QA Datasets.** We provide a comprehensive evaluation of HyCo$_2$ against compression and retrieval methods across seven downstream tasks, as shown in Table 1. The RAG baseline, which leverages full retrieved context without compression, yields notable improvements over the vanilla LLM setting. Among the evaluated compression techniques, HyCo$_2$ consistently achieves the highest average EM across all three LLMs, outperforming EXIT and xRAG while retaining more relevant information. Importantly, HyCo$_2$ introduces only 168M parameters during inference, significantly fewer than EXIT (4B) and xRAG (7B) and reduces token usage by 88.8% on average, with minimal or no performance loss. On certain datasets (WQ, CWQ), HyCo$_2$ even surpasses the uncompressed RAG setting (+0.7% and +10% EM with 89.1% fewer tokens using Mistral-7B).

We further observe that for advanced models like LLaMA3.1 and Qwen2.5, RAG may underperform vanilla models on tasks that rely heavily on Wikipedia (WQ, CWQ, HQA), likely due to redundancy or conflict between retrieved content and pre-trained knowledge. This is supported by the superior performance of most compression-based methods. Finally, unlike xRAG's single-token soft compression, HyCo$_2$ demonstrates stronger multi-document reasoning capabilities, particularly on complex tasks such as HQA and 2WIKI.

**LongBench.** We evaluate HyCo$_2$ on the LongBench benchmark (Bai et al., 2024), which comprises diverse long-context understanding tasks spanning question answering, summarization, classification, and retrieval. As shown in Table 2, HyCo$_2$ consistently outperforms other compression baselines under a 2k-token constraint. Compared to LLMLingua2 and EXIT, HyCo$_2$ achieves significantly higher average performance (+5.68 and +11.37, respectively), particularly on knowledge-intensive tasks such as GovReport and Qasper. Notably, the EXIT evaluation on 4×H100 GPUs remains incomplete due to prolonged runtime (currently exceeding 45 hours), highlighting its computational inefficiency. In contrast, HyCo$_2$ offers a balanced trade-off between compression and performance, exhibiting robustness across tasks with significantly lower inference overhead.

Table 1: Performance comparison between our HyCo$_2$ and other methods ( Uncompressed , Hard and Soft compression) on seven downstream tasks. Percentages in brackets denote the relative improvement over the non-retrieval (Vanilla) setting in average performance (Avg.) and RAG setting in context length. The best results are in **bold** and the underline indicates the dataset is IID. LLMs are frozen during the experiments and retrieved documents are set the same for different methods.

| | Methods | Addit. Size ↓ | # Context Length ↓ | Open-Domain QA (EM ↑) | | | | | Multihop QA (EM ↑) | | |
| | | | | NQ | TQA | WQ | PQA | CWQ | HQA | 2WIKI | Avg. |
|---|---|---|---|---|---|---|---|---|---|---|---|
| Mistral-7B-Ins.-v0.2 | Vanilla | - | 0 (↓**100%**) | 34.4 | 59.4 | 42.2 | 21.3 | 48.0 | 26.4 | 36.7 | 38.34 (0.0%) |
| | RAG | - | 466.9 (**100%**) | 54.4 | 71.3 | 45.1 | 67.0 | 45.7 | 29.5 | 40.6 | 50.51 (↑ 31.7%) |
| | TF-IDF | - | 64 (↓ 86.3%) | 34.4 | 60.6 | 38.8 | 30.7 | 43.3 | 23.0 | 39.6 | 38.63 (↑ 0.8%) |
| | LongLLMLingua (Jiang et al., 2024) | 7B | 131.2 (↓ 71.9%) | 39.5 | 64.3 | 39.3 | 44.3 | 49.0 | 24.9 | 39.0 | 42.90 (↑ 11.9%) |
| | LLMLingua2 (Pan et al., 2024) | 561M | 114.2 (↓ 75.5%) | 38.1 | 62.5 | 41.1 | 43.7 | 45.0 | 25.5 | 38.9 | 42.11 (↑ 9.8%) |
| | EXIT (Hwang et al., 2024) | 4B | 83.7 (↓ 82.0%) | 41.9 | 65.4 | 43.0 | 47.3 | 49.0 | 27.2 | 39.9 | 44.81 (↑ 16.8%) |
| | ICAE (Ge et al., 2023b) | - | 128 (↓ 73.6%) | 20.6 | 57.8 | 39.2 | 41.7 | 46.3 | 20.8 | 24.5 | 35.84 (↓ 6.5%) |
| | COCOM-Light (Rau et al., 2024) | 561M | 115.7 (↓ 75.2%) | 40.3 | 65.7 | 42.9 | 44.7 | 48.7 | 25.9 | 36.4 | 43.51 (↑ 13.4%) |
| | xRAG (Cheng et al., 2024) | 7B + 35M | 3 (↓ 99.4%) | 37.2 | 65.5 | 43.4 | 39.3 | 47.7 | 22.0 | 25.9 | 40.14 (↑ 4.7%) |
| | **HyCo$_2$** (ours) | 168M | 50.7 (↓ 89.1%) | 39.6 | **66.0** | **45.4** | 45.7 | **50.3** | 27.5 | **40.2** | 44.96 (↑ 17.3%) |
| LLaMA-3.1-8B-Ins. | Vanilla | - | 0 (↓**100%**) | 38.0 | 67.0 | 50.6 | 33.0 | 49.0 | 27.7 | 31.9 | 42.46 (0.0%) |
| | RAG | - | 466.9 (**100%**) | 52.6 | 71.0 | 40.4 | 56.0 | 40.0 | 27.3 | 34.0 | 46.51 (↑ 9.5%) |
| | TF-IDF | - | 64 (↓ 86.3%) | 37.0 | 64.7 | 35.4 | 27.0 | 41.3 | 23.0 | 31.3 | 37.10 (↓ 12.6%) |
| | LongLLMLingua (Jiang et al., 2024) | 7B | 131.2 (↓ 71.9%) | 38.1 | 66.4 | 34.3 | 40.3 | 49.0 | 25.7 | 32.4 | 40.89 (↓ 3.7%) |
| | LLMLingua2 (Pan et al., 2024) | 561M | 114.2 (↓ 75.5%) | 37.4 | 65.2 | 35.8 | 39.7 | 42.0 | 24.9 | 31.5 | 39.50 (↓ 7.0%) |
| | EXIT (Hwang et al., 2024) | 4B | 83.7 (↓ 82.0%) | 41.5 | 66.5 | 40.1 | 47.3 | 48.7 | 29.9 | 33.1 | 43.87 (↑ 3.3%) |
| | xRAG (Cheng et al., 2024) | 7B + 35M | 3 (↓ 99.4%) | 35.6 | 64.8 | 40.0 | 34.7 | 49.0 | 24.1 | 28.1 | 39.47 (↓ 7.0%) |
| | **HyCo$_2$** (ours) | 168M | 52.1 (↓ 88.8%) | 39.3 | **67.1** | 40.8 | 46.7 | **49.7** | **30.5** | **33.6** | 43.96 (↑ 3.5%) |
| Qwen-2.5-7B-Ins. | Vanilla | - | 0 (↓**100%**) | 29.6 | 55.1 | 39.1 | 23.7 | 44.7 | 25.5 | 31.2 | 35.56 (0.0%) |
| | RAG | - | 466.9 (**100%**) | 51.9 | 69.6 | 40.9 | 56.0 | 35.7 | 21.3 | 35.5 | 44.41 (↑ 24.9%) |
| | TF-IDF | - | 64 (↓ 86.3%) | 28.9 | 56.2 | 35.3 | 11.7 | 37.3 | 20.0 | 31.8 | 31.60 (↓ 11.1%) |
| | LongLLMLingua (Jiang et al., 2024) | 7B | 131.2 (↓ 71.9%) | 33.4 | 59.8 | 35.3 | 43.7 | 38.7 | 21.3 | 31.7 | 37.70 (↑ 6.0%) |
| | LLMLingua2 (Pan et al., 2024) | 561M | 114.2 (↓ 75.5%) | 30.9 | 56.5 | 34.2 | 12.7 | 45.3 | 20.2 | 31.2 | 31.40 (↓ 11.7%) |
| | EXIT (Hwang et al., 2024) | 4B | 83.7 (↓ 82.0%) | 37.2 | 59.4 | 40.3 | **51.7** | 45.3 | **26.7** | 32.7 | 41.90 (↑ 17.8%) |
| | xRAG (Cheng et al., 2024) | 7B + 35M | 3 (↓ 99.4%) | 27.9 | 53.7 | 39.7 | 23.7 | 46.0 | 23.1 | 27.9 | 34.57 (↓ 2.8%) |
| | **HyCo$_2$** (ours) | 168M | 53.4 (↓ 88.6%) | 34.6 | **60.2** | **43.1** | 50.7 | **46.3** | 26.2 | **33.8** | 42.11 (↑ 18.4%) |

Table 2: Performance comparison on LongBench using LLaMA3-8B-Instruct within 2k tokens.

| Method | Single-Document QA | | | Multi-Document QA | | | Summarization | | | Few-shot Learning | | | Synthetic | | Code | | Avg. |
| | NrtvQA | Qasper | MF-en | HotpotQA | 2WikiMQA | Musique | GovReport | QMSum | MultiNews | TREC | TriviaQA | SAMSum | PCount | PRe | Lcc | RB-P | |
|---|---|---|---|---|---|---|---|---|---|---|---|---|---|---|---|---|---|
| Vanilla | 29.86 | 45.37 | 55.12 | 55.22 | 44.87 | 31.77 | 35.24 | 25.53 | 27.23 | 72.5 | 91.64 | 43.64 | 7.38 | 99.5 | 63.23 | 56.63 | 49.69 |
| LongLLMLingua | 12.8 | 19.83 | 30.57 | 22.56 | 21.11 | 14.73 | 25.46 | 21.64 | 21.8 | 3.25 | 63.91 | 21.1 | 4.0 | 8.5 | 22.5 | 43.32 | 22.32 |
| LLMLingua2 | 2.85 | 28.92 | 23.82 | 4.37 | 9.7 | 1.56 | 26.79 | 15.02 | 26.91 | 11.5 | 33.17 | 15.27 | 0.0 | 5.29 | 35.04 | 25.06 | 16.23 |
| EXIT | 4.09 | 7.84 | 15.15 | 6.85 | 6.07 | 5.14 | 8.75 | 13.77 | 4.47 | - | 0.0 | 7.08 | 0.0 | 0.0 | 18.78 | 23.18 | 8.21 |
| **HyCo$_2$** | 4.87 | 38.21 | 28.21 | 11.56 | 18.32 | 6.21 | 29.88 | 18.47 | 26.18 | 19.34 | 45.84 | 20.31 | 0.0 | 12.54 | 41.27 | 31.25 | 21.94 |

## 3.3 ANALYSIS

**Information Preservation.** To assess the ability to preserve input information, we prompt the target LLM to reconstruct the original context from the compressed inputs (prompt templates are detailed in Appendix E). This evaluation focuses on xRAG-based methods, excluding hard compression baselines, as the latter are fully interpretable and do not generate novel content. We adopt four metrics—BERTScore, Information Loss, ROUGE-L, and Readability—with detailed definitions in Appendix C.4. As shown in Figure 4(a), HyCo$_2$ achieves the best overall reconstruction quality on both TQA and 2Wiki. Specifically, it yields an average BERTScore F1 gain of 0.05, a 0.5-point reduction in information loss, and consistent improvements in ROUGE-L and readability, highlighting its strength in retaining both global semantics and fine-grained details.

**Robustness.** We evaluate robustness under increasing input lengths by varying the number of retrieved documents ($K \in 1, 3, 5, 8, 10$), as shown in Figure 4(b). When $K \leq 5$ (i.e., context length under 1k), HyCo$_2$ maintains stable EM performance, comparable to or better than the RAG baseline. In contrast, competing method including xRAG begin to degrade at $K \geq 3$, with xRAG peaking only at $K = 1$, aligning with prior observations in Cheng et al. (2024). Despite the general decline across all methods as $K$ increases, HyCo$_2$ exhibits significantly slower degradation. Notably, at $K = 10$,

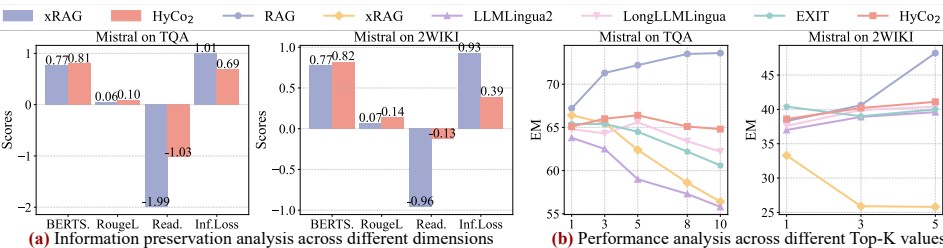

Figure 4: We employ Mistral-7B to investigate two aspects: (a) a four-dimensional comparison of information preservation between HyCo$_2$ and xRAG following context compression and reconstruction, and (b) the performance trends of various compression methods as context length increases. BERTScore measures semantic similarity, Information Loss measures the entropy value of discarded information, while Readability and ROUGE-L evaluate the quality of the reconstructed context.

its EM score drops by just 1.2 points—demonstrating superior robustness in handling long contexts. These findings underscore persistent challenges in context compression for long texts and emphasize the advantages of our hybrid compression strategy.

**Efficiency and Memory.** We utilize *Torch Profiler* to evaluate the efficiency across different methods on various datasets, measuring CPU time (s), CUDA time (s), computations (GFLOPs), and peak GPU memory usage (GB). All experiments are conducted using Mistral-7B and LLaMA3.1-8B in BFloat16 inference mode on a single A100 GPU, with a batch size of 1 and a fixed output length of 30. As shown in Table 3, HyCo$_2$ achieves the best performance in terms of CPU time (0.572 s) and CUDA time (0.187 s). It also attains the lowest peak memory usage (14.56 GB), saving approximately 50% GPU memory compared to xRAG, which is consistent with the additional memory overhead from xRAG's embedding model. In terms of GFLOPs (312.73), HyCo$_2$ outperforms xRAG and LLMLingua2, while remaining significantly more efficient than EXIT. Notably, although xRAG has the lowest GFLOPs, it exhibited the highest memory consumption. In contrast, EXIT incurres the highest computational and time costs among all methods.

### 3.4 ABLATION STUDIES

**Components Analysis.** Table 5 reports an ablation study assessing the contribution of each component in HyCo2. Removing the instruction-conditioned cross-attention notably degrades performance, confirming the importance of instruction signals in guiding the compressor to retain task-relevant content. For objective functions, the self-distillation loss $\mathcal{L}_{kl}$ surpasses standard language modeling loss $\mathcal{L}_{nll}$, as it better aligns the compressor with informative teacher representations. Furthermore, both the pretraining and instruction tuning stages are indispensable, jointly enhancing overall performance and validating our staged training paradigm.

**Effects of Hybrid Adapter.** We compare several query-based and pooling-based strategies for global context compression, including learnable query tokens (QFormer), pooling projection (AdaPool), and a single learnable token (One Token), with results shown in Table 5 (Query Type section). While One Token is inadequate for retaining nuanced semantics, AdaPool better preserves global context. QFormer, despite its theoretical advantage in instruction-aware retrieval, is found to underperform in practice. Notably, the combination of learnable queries with pooling yields the best results, which aligns with findings from prior work (Zhang et al., 2024b).

**Hyperparameter Selection.** As shown in the bottom of the Table 5, our hyperparameter analysis for $N_L$ identified a trade-off between task-specific capability and general efficiency. While a larger $N_L$ (e.g., 32) enhanced performance on complex multi-hop QA, it offered diminishing returns overall and significantly lowered the compression ratio. Conversely, a smaller $N_L$ (e.g., 8) was broadly detrimental. We thus determined $N_L = 16$ to be the optimal setting, providing a robust balance across diverse task complexities. A similar rationale was applied to $k$.

**Impact of Alternating Training.** We evaluate the alternating training strategy against a conventional end-to-end (E2E) baseline and observe an average performance drop of 2% under E2E training. Omitting Stage 2 pretraining further degrades performance, underscoring the necessity of initializing the

Table 3: Comparison of context compression methods about efficiency and memory usage.

| Method | CPU Time (s) | CUDA Time (s) | GFLOPs | Peak Mem. (GB) |
|---|---|---|---|---|
| *Mistral-7B-Instruct-v0.2 on TQA* | | | | |
| xRAG | 0.716 | 0.249 | **253.25** | 27.05 |
| LLMLingua2 | 1.037 | 0.418 | 264.77 | 16.60 |
| EXIT | 2.495 | 0.820 | 1624.37 | 20.43 |
| **HyCo$_2$** (ours) | **0.572** | **0.187** | 312.73 | **14.56** |
| *Mistral-7B-Instruct-v0.2 on 2WIKI* | | | | |
| xRAG | 0.787 | 0.252 | **181.89** | 27.06 |
| LLMLingua2 | 1.031 | 0.408 | 192.58 | 16.60 |
| EXIT | 1.639 | 0.626 | 1142.99 | 20.41 |
| **HyCo$_2$** (ours) | **0.672** | **0.197** | 228.50 | **14.78** |
| *LLaMA3.1-8B-Instruct on TQA* | | | | |
| xRAG | 0.591 | 0.248 | 251.99 | 28.52 |
| LLMLingua2 | 0.656 | 0.178 | **242.95** | 18.50 |
| EXIT | 1.456 | 0.665 | 1602.54 | 21.90 |
| **HyCo$_2$** (ours) | **0.324** | **0.136** | 288.05 | **16.92** |
| *LLaMA3.1-8B-Instruct on 2WIKI* | | | | |
| xRAG | 0.575 | 0.228 | **180.02** | 28.53 |
| LLMLingua2 | 0.854 | 0.234 | 188.04 | 18.80 |
| EXIT | 0.916 | 0.395 | 962.47 | 21.88 |
| **HyCo$_2$** (ours) | **0.334** | **0.126** | 211.53 | **17.38** |

Table 4: Ablation study on router design.

| Method | NQ | TQA | HQA | 2WIKI |
|---|---|---|---|---|
| **HyCo$_2$** | **39.6** | **66.0** | **27.5** | **40.2** |
| Concat | 36.7 | 64.8 | 26.9 | 39.1 |
| w/o soft | 36.0 | 64.2 | 26.3 | 38.7 |
| Fixed-threshold | 34.2 | 62.6 | 25.9 | 37.9 |

Table 5: Results of Ablation Studies. The row with a `gray background` indicates our default setting. The backbone model is Mistral-7B.

| Method | NQ | TQA | HQA | 2WIKI |
|---|---|---|---|---|
| HyCo$_2$ | 39.6 | 66.0 | 27.5 | 40.2 |
| w/o Ins. | 38.8 (-0.8) | 65.5 (-0.5) | 26.1 (-1.4) | 38.6 (-1.6) |
| w/o $\mathcal{L}_{nll}$ | 37.7 (-1.9) | 63.9 (-2.1) | 26.7 (-0.8) | 41.4 (+1.2) |
| w/o $\mathcal{L}_{kl}$ | 35.2 (-4.4) | 62.6 (-3.4) | 26.4 (-1.1) | 38.8 (-1.4) |
| w/o Pretrain | 34.2 (-5.4) | 59.4 (-6.6) | 25.0 (-2.5) | 38.2 (-2.0) |
| w/o Finetune | 33.1 (-6.5) | 60.7 (-5.3) | 25.6 (-1.9) | 39.4 (-0.8) |
| *Query Type* | | | | |
| One Token | 33.5 (-6.1) | 60.0 (-6.0) | 25.4 (-2.1) | 37.1 (-3.1) |
| AdaPool | 36.4 (-3.2) | 63.0 (-3.0) | 28.0 (+0.5) | 38.9 (-1.3) |
| QFormer | 34.7 (-4.9) | 63.9 (-2.1) | 26.8 (-0.7) | 37.7 (-2.5) |
| Hybrid | 39.6 | 66.0 | 27.5 | 40.2 |
| *Training strategies* | | | | |
| E2E | 36.8 (-2.8) | 62.8 (-3.2) | 26.4 (-1.1) | 38.3 (-1.9) |
| w/o Stage 2 | 36.3 (-3.3) | 62.0 (-4.0) | 25.6 (-1.9) | 37.8 (-2.4) |
| w/o Global | 29.7 (-9.9) | 55.7 (-10.3) | 22.4 (-5.1) | 35.0 (-5.2) |
| w/o Local | 33.6 (-6.0) | 60.5 (-5.5) | 24.8 (-2.7) | 37.9 (-2.3) |
| Alternating | 39.6 | 66.0 | 27.5 | 40.2 |
| w/o Stage 2 | 37.8 (-1.8) | 64.1 (-1.9) | 27.1 (-0.4) | 39.3 (-0.9) |
| w/o Global | 33.2 (-6.4) | 58.8 (-7.2) | 24.7 (-2.8) | 37.3 (-2.9) |
| w/o Local | 35.4 (-4.2) | 63.6 (-2.4) | 26.6 (-0.9) | 38.9 (-1.3) |
| *Local k%* | | | | |
| 0 | 35.4 | 63.6 | 26.6 | 38.9 |
| 5 | 37.5 | 64.8 | 27.0 | 39.6 |
| 10 | 39.6 | 66.0 | 27.5 | 40.2 |
| 20 | 40.3 | 66.8 | 27.6 | 40.5 |
| *Number of Query $N_L$* | | | | |
| 8 | 37.8 | 65.1 | 26.7 | 38.4 |
| 16 | 39.6 | 66.0 | 27.5 | 40.2 |
| 32 | 40.4 | 66.9 | 28.0 | 40.9 |

local compressor with token importance signals. Isolated training of local or global modules reveals that local compression alone fails under both E2E and alternating setups due to excessive information loss. Conversely, global-only training yields better robustness, suggesting global semantics are more crucial for downstream reasoning. These results collectively highlight the utility of our alternating training regime for effective compression learning.

**Design of Router.** As shown in Table 4, removing the QFormer and directly concatenating compressed outputs results in a 2.9% EM drop on NQ, underscoring the benefit of task-adaptive fusion. Omitting soft compression and relying solely on the router degrades performance by 3.6%, confirming the complementarity of global semantics and local salience. Replacing the learnable MLP gate with a fixed threshold further reduces accuracy, particularly on instruction-heavy tasks, validating the necessity of dynamic, learnable routing.

# 4 RELATED WORK

Context compression aims to reduce the input length of LLMs while preserving essential information. Existing methods typically fall into two paradigms: *hard compression* and *soft compression*.

**Hard compression** selectively retains or paraphrases input tokens in natural language form to minimize input length. Filtering-based methods such as SelectiveContext (Li et al., 2023b) and LLMLingua (Jiang et al., 2023b) leverage token-level metrics (e.g., self-information or external LM scores) to remove low-utility content. Extensions like LongLLMLingua (Jiang et al., 2024) and TACO-RL (Shandilya et al., 2024) further adapt filtering to long contexts or reinforcement learning objectives. While effective, these methods may compromise syntactic coherence, introduce grammatical errors, and exhibit limited robustness across LLM variants. Paraphrasing-based approaches, including Nano-Capsulator (Chuang et al., 2024), utilize fine-tuned models to rewrite inputs into more compact forms. Although they often improve compression quality, the generative process incurs substantial computational overhead (Liao et al., 2025a).

**Soft compression** encodes inputs into dense, continuous representations (e.g., embeddings or memory slots) to bypass the quadratic cost of attention. Approaches such as GIST (Mu et al., 2023),

AutoCompressor (Chevalier et al., 2023), and ICAE (Ge et al., 2023b) reduce context length by modifying attention mechanisms or introducing latent memory representations. While these methods achieve higher compression rates and scalability, they often disrupt input structure, lose fine-grained details, and diminish interpretability (Deng et al., 2024). Additionally, projection-based strategies (UniICL (Gao et al., 2024), xRAG (Cheng et al., 2024)) utilize MLPs to map input into fixed-size latent vectors, albeit at the expense of information fidelity and increased memory consumption.

While some recent efforts explore latent-only hybrid compression (Liu et al., 2025b), our proposed $HyCo_2$ explicitly integrates local token retention with global semantic abstraction. Importantly, $HyCo_2$ introduces minimal additional parameters and avoids reliance on external models.

## 5 CONCLUSION

In this work, we propose $HyCo_2$, a novel framework that enables efficient long-context inference in large language models (LLMs) by balancing local detail preservation and global semantic abstraction. $HyCo_2$ combines hard compression, which retains fine-grained, instruction-relevant local information, with soft compression, which captures high-level contextual semantics, thereby reducing token overhead without sacrificing critical content. We adopt an alternating training scheme, pretraining the global and local compression modules with paraphrasing and completion tasks, respectively, and subsequently performing instruction tuning for downstream alignment. Experimental results show that $HyCo_2$ substantially improves performance on knowledge-intensive benchmarks, including open-domain and multi-hop QA. Overall, $HyCo_2$ offers a lightweight, effective, and generalizable solution for long-context reasoning in LLMs.

## ETHICS STATEMENT

Our approach does not introduce ethical concerns. The datasets and models we used are public, and there are no privacy issues.

## REPRODUCIBILITY STATEMENT

In this work, we use open-source LLMs and publicly available datasets to conduct our experiments. To ensure reproducibility, we provide the implementation details in the Section 3.1 and the Appendix C and the full code in https://anonymous.4open.science/r/HyCo2.

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

## A  CASE STUDY FOR CRITICAL INTERPLAY BETWEEN LOCAL DETAIL PRESERVATION AND GLOBAL SEMANTIC COMPLETENESS

Consider the document containing the entries in Table 1 (b):

**Global Semantic Completeness** is essential for accurate query interpretation in this context. It entails recognizing that the document discusses **two distinct individuals** named *George Rankin*, rather than a single person. A compression method that conflates these entities or represents only the first instance fails to preserve the document's overarching semantic structure. Specifically, it would neglect the ambiguity inherent in the term "George Rankin," omitting the fact that multiple, disambiguated profiles are present. The compressed representation must therefore retain the **core semantic meaning**: that "George Rankin" refers to more than one person, each associated with a unique set of attributes.

**Local Detail Preservation**, by contrast, concerns the retention of fine-grained, entity-specific information. For *Sir George Claus Rankin*, this includes his full name, honorific title ("Sir"), professional role ("British judge in India"), and lifespan. For *Major General George James Rankin*, the critical local details include his full name, military rank ("Major General"), professional roles ("Australian soldier and politician"), and service record. If the compression process omits these elements, such as the occupations or titles, it undermines the factual integrity of the representation, even if the presence of multiple entities is correctly preserved.

Accordingly, an effective compression method must satisfy both criteria. It must maintain global semantic completeness by encoding the presence of multiple individuals named "George Rankin," and simultaneously ensure local detail preservation by retaining the specific identifiers that distinguish them. A compressed output that enables a system to generate the response, "The document refers to two individuals: *Sir George Claus Rankin, a British judge in India, and Major General George James Rankin, an Australian soldier and politician*," would exemplify successful integration of these principles. This case underscores that failure in either global disambiguation or local specificity significantly compromises the utility of compressed representations for downstream reasoning and information retrieval tasks.

## B  ALTERNATING TRAINING STRATEGY

### B.1  PARAPHRASE PRETRAINING

In Stage 1, the objective is to train the hybrid compressor to align the soft-gated global token with the original context $x$'s global semantics. Specifically, the LLM utilizes natural language instructions $\mathbf{X}_{\texttt{paraphrase}}$[1] to generate context, aiming to reconstruct the original context. The optimization objective is defined by the following formula:

$$\mathcal{L}_{\text{nll}} = -\sum_{i=1} \log p_{\phi}(x_i | \mathcal{G}(\mathcal{F}_{\phi}(\boldsymbol{x})), \mathbf{X}_{\texttt{paraphrase}}, x_{<i}) \tag{5}$$

---

[1]To maintain diversity, we sample from an instruction pool, which could be found in Appendix E.1.

where $p_\phi$ is given by the softmax distribution of LLM $\mathcal{F}_\phi$, $\mathcal{F}_\phi(\boldsymbol{x})$ is the context feature encoded by Encoder (LLM itself), $\mathcal{G}$ is a learned gating network and $x_{<i}$ denotes the context token before current prediction token $x_i$, achieved by casual attention mask in auto-regressive LMs.

### B.2 COMPLETION PRETRAINING

In Stage 2, the context $\boldsymbol{x}$ from RedPajama-Data-V2 is randomly partitioned into two segments: $a$ and $b$. Segment $a$ functions as the context, while segment $b$ is the target for prediction. By minimizing the negative log-likelihood $\mathcal{L}_{\text{nll}}$ of predicting segment $b$ given the compressed context of $a$ (formed using the local classification layer), the model is trained to preserve the key information from context $a$ necessary to generate $b$ using instructions $\mathbf{X}_{\texttt{completion}}$. The optimization objective is:

$$\mathcal{L}_{\text{nll}} = -\sum_{i=1} \log p_\phi(b_i | \mathcal{H}(\mathcal{F}_\phi(a)), \mathcal{G}(\mathcal{F}_\phi(a)), \mathbf{X}_{\texttt{completion}}, b_{<i}) \tag{6}$$

where $\mathcal{H}$ is the local classification layer for keeping top $k\%$ tokens.

### B.3 INSTRUCTION TUNING

In Stage 3, we utilize triplets $(\boldsymbol{q}, \boldsymbol{x}, \boldsymbol{y})$ where $\boldsymbol{q}$ is the question, $\boldsymbol{x}$ is the context (retrieved documents or long input), and $\boldsymbol{y}$ is the output answer. On one hand, we employ a language modeling objective, consistent with the first two stages, to train the model to generate the correct output $\boldsymbol{y}$ based on task-specific instructions and the provided context $\boldsymbol{x}$:

$$\mathcal{L}_{\text{nll}} = -\sum_{i=1} \log p_\phi(\boldsymbol{y}_i | \mathcal{H}(\mathcal{F}_\phi(\boldsymbol{x})), \mathcal{G}(\mathcal{F}_\phi(\boldsymbol{x})), \boldsymbol{q}, \boldsymbol{y}_{<i}) \tag{7}$$

On the other hand, we incorporate self-distillation (Cheng et al., 2024), treating the RAG model as the teacher and HyCo$_2$ as the student to transfer knowledge. This process trains HyCo$_2$ to simulate the RAG model's proficiency in handling complete, uncompressed documents, thereby facilitating the learning of more effective compressed representations. Specifically, for the base language model $\mathcal{F}_\phi$, which receives either the uncompressed context $\boldsymbol{x}$ (from the teacher RAG model) or the compressed representation $(\mathcal{H}(\mathcal{F}_\phi(\boldsymbol{x})), \mathcal{G}(\mathcal{F}_\phi(\boldsymbol{x})))$ (from HyCo$_2$), the objective is to minimize the divergence between the two resulting output distributions. This divergence is measured using the Kullback-Leibler (KL) divergence:

$$\mathcal{L}_{\text{kl}} = \mathcal{D}_{\text{KL}}(p_\phi(\boldsymbol{y}|\boldsymbol{x}, \boldsymbol{q}) \,||\, p_\phi(\boldsymbol{y}|\mathcal{H}(\mathcal{F}_\phi(\boldsymbol{x})), \mathcal{G}(\mathcal{F}_\phi(\boldsymbol{x})), \boldsymbol{q})) \tag{8}$$

The final loss is the linear combination controlled by a hyperparameter: $\mathcal{L}_{\text{nll}} + \alpha \mathcal{L}_{\text{kl}}$.

## C EXPERIMANTAL SETTINGS

### C.1 DATASETS

#### C.1.1 DETAILS FOR PRETRAINING DATASET

For Paraphrase Pretraining, we construct training instances derived from the retrieval corpus $\mathbb{D}$. Each instance involves employing natural language instructions to prompt the LLM to generate a paraphrase or description (Cheng et al., 2024).

For Completion Pretraining, we randomly split documents from the RedPajama-Data-V2 (Weber et al., 2024) "2023-06" snapshot into two segments, where the length of the second segment is randomly sampled from the range [5, 100] to simulate realistic generation lengths.

#### C.1.2 DETAILS FOR INSTRUCTION TUNING DATASET

We utilize the same instruction fine-tuning dataset as xRAG (Cheng et al., 2024). Table 6 provides a summary, and Table 7 offers detailed information about each subtask within the dataset. For question-answering tasks originally lacking explicit context, we employ Contriever (Izacard et al., 2021) to perform retrieval on the corpus $\mathbb{D}$, selecting the top-10 documents to serve as context.

Table 6: Overall statistics of Instruction Tuning dataset.

| Task Type | # Involved datasets | # Train | # Prompt | # Label |
|---|---|---|---|---|
| Reading Comprehension | 7 | 488,344 | 447.62 | 30.34 |
| Summarization | 3 | 81,821 | 483.49 | 53.29 |
| Open Domain QA | 7 | 385,173 | 203.55 | 20.09 |

Table 7: Detailed data statistics for our Context-aware Instruction Tuning Dataset.

| Task Type | Dataset | # Train | # Prompt Len | # Label Len |
|---|---|---|---|---|
| Reading Comprehension | CoQA (Reddy et al., 2019) | 7101 | 617.98 | 77.75 |
| | DROP (Dua et al., 2019) | 76098 | 356.06 | 3.86 |
| | NarrativeQA (Kočiský et al., 2017) | 32747 | 702.39 | 7.86 |
| | PubMedQA (Jin et al., 2019) | 1000 | 397.91 | 65.4 |
| | QuAIL (Rogers et al., 2020) | 10246 | 512.9 | 2.0 |
| | SQuAD v2 (Rajpurkar et al., 2018) | 130319 | 214.54 | 6.87 |
| | PwC (Ge et al., 2023a) | 241564 | 571.35 | 53.07 |
| Open Domain QA | NQ (Kwiatkowski et al., 2019) | 87925 | 203.62 | 5.976 |
| | TriviaQA (Joshi et al., 2017) | 78785 | 216.1 | 6.49 |
| | CommonsenseQA (Talmor et al., 2019) | 9741 | 223.64 | 2.0 |
| | WikiQA (Yang et al., 2015) | 1040 | 192.89 | 40.79 |
| | YahooQA | 87358 | 196.56 | 56.7 |
| | FreebaseQA (Jiang et al., 2019) | 20353 | 218.49 | 4.87 |
| | MSMarco (Bajaj et al., 2018) | 99994 | 194.82 | 15.91 |
| Summarization | CNN/DM (See et al., 2017) | 100000 | 616.99 | 63.37 |
| | SamSum (Gliwa et al., 2019) | 14731 | 187.87 | 29.12 |
| | DialogSum (Chen et al., 2021) | 12460 | 247 | 37.61 |

### C.1.3 EVALUATION DATASET

To ensure a comprehensive evaluation, we assess our method using the following 5 Open-Domain QA and 2 multihop QA:

- **NaturalQuestions (NQ)** (Kwiatkowski et al., 2019) contains questions corresponding to Google search queries. The open-domain version of this dataset is obtained by discarding answers with more than 5 tokens, each accompanied by a Wikipedia article containing the answer.

- **TriviaQA (TQA)** (Joshi et al., 2017) contains questions gathered from trivia and quiz-league websites. The unfiltered version of TriviaQA is used for open-domain question answering, each question is accompanied by pages from web and Wikipedia searches that may contain the answer.

- **WebQuestions (WQ)** (Berant et al., 2013) contains questions from web queries matched to corresponding entries in FreeBase.

- **PopQA (PQA)** (Mallen et al., 2023) focuses on factual question answering, posing challenges that test the model's ability to recall precise knowledge and navigate ambiguities in entity representation.

- **ComplexWebQuestions (CWQ)** (Talmor & Berant, 2018) entails answering complex, multi-step questions sourced from the web, further challenging the model's capacity to retrieve and reason over extensive web content.

- **2WikiMultihopQA (2WIKI)** (Ho et al., 2020) is designed to evaluate a model's capability in multi-hop reasoning by synthesizing information from multiple Wikipedia passages.

- **HotpotQA (HQA)** (Yang et al., 2018) similarly targets multi-hop reasoning, requiring models to amalgamate information from various contexts to answer a single query.

## C.2 IMPLEMENTATIONS

Our implementations are based on Huggingface Transformers v4.45.2 (Wolf et al., 2020) using PyTorch v2.3.0 (Paszke, 2019) and deepspeed[2] v0.14.0. All experiments were conducted on 8 A100 NVIDIA GPUs, each equipped with 80GB of memory. In Table 8 and Table 9, we list the hyperparameters for Pretraining and Instruction Tuning.

Table 8: Hyperparameters for Pretraining.

| Hyperparameter | Assignment |
|---|---|
| query tokens number | 16 |
| k% | 10% |
| optimizer | AdamW |
| learning rate | 1e-4 |
| lr scheduler type | linear |
| warmup ratio | 0.03 |
| weight decay | 0.0 |
| epochs | 1 |
| flash attention | True |
| batch size | 4 |
| gradient accumulation steps | 4 |
| num GPUs | 8 |
| max sequence length | 2048 |
| max train samples | 1,000,000 |

Table 9: Hyperparameters for Instruction Tuning.

| Hyperparameter | Assignment |
|---|---|
| query tokens number | 16 |
| k% | 10% |
| optimizer | AdamW |
| learning rate | 2e-5 |
| lr scheduler type | linear |
| warmup ratio | 0.03 |
| weight decay | 0.0 |
| epochs | 1 |
| KL $\alpha$ | 2.0 |
| KL temperature | 1.0 |
| flash attention | True |
| batch size | 4 |
| gradient accumulation steps | 4 |
| num GPUs | 8 |
| max sequence length | 4096 |
| max train samples | 955,338 |

## C.3 BASELINES

Since the LLM in our method remains frozen, the selected baselines must support plug-and-play functionality without requiring any alteration to the LLM's parameters (Mu et al., 2023). Accordingly, we focus on three categories of baselines: 1) Uncompressed: **Vanilla**: Represents the original LLM, which generates answers directly without utilizing any external information. **RAG**: Appends the top retrieved documents to the LLM's input prompts, explicitly instructing the model to reference them when generating answers. 2) Hard Compression: **TF-IDF**: Performs topic-based discrete compression using term frequency-inverse document frequency. **LongLLMLingua** (Jiang et al., 2024) uses LLaMA2-7B-chat for token-level extraction with a 0.4 dynamic compression rate. **LLMLingua2** (Pan et al., 2024): A RoBERTa model trained on compressed data distilled from GPT-4. **EXIT** (Hwang et al., 2024): Adaptively classifies and extracts contextually dependent sentences from retrieved documents. Soft Compression: **xRAG** (Cheng et al., 2024): Uses MLPs to project the last token representation of the *top-1* document.

## C.4 INFORMATION PRESERVATION METRICS

**BERTScore** is a metric used to evaluate the semantic similarity between a compressed text and its source. Unlike traditional metrics that rely on surface-level n-gram matching, BERTScore leverages contextual embeddings from models like BERT to compute similarity at the semantic level.

**Information Loss** quantifies the amount of information from the original text that is not successfully retained in the compressed text. A lower information loss indicates a more effective compression method in terms of preserving content. Information quantity can be measured using the concept of Entropy ($H$). Higher entropy generally corresponds to higher information content. The information loss is defined as the difference between the information content of the original text $x$ and the compressed text $\hat{x}$, i.e., $H_x - H_{\hat{x}}$.

---

[2]https://github.com/microsoft/DeepSpeed

**ROUGE** is a widely used set of metrics for evaluating the quality of automatically generated text summaries by comparing them to reference summaries (in this context, comparing the compressed text to the source or a gold standard summary derived from it). It primarily measures the overlap of units like n-grams or sequences between the compressed text and the original.

**Readability** assesses how easy a text is to read and understand. For compressed text, it measures the linguistic fluency and naturalness of the resulting output. Readability can be estimated using automated readability formulas, such as the Flesch Reading Ease score. These formulas typically consider factors like sentence length and the number of syllables per word to produce a score indicating reading difficulty.

## D  LIMITATIONS

While our proposed $HyCo_2$ demonstrates significant improvements in balancing local and global information retention for large language models (LLMs), several limitations warrant further investigation.

**Performance on Minimal Contexts (Top-1 Document)**: When processing only the top-1 retrieved document, particularly on certain IID datasets such as Natural Questions (NQ) and TriviaQA (TQA), HyCo2's performance may not consistently surpass xRAG. The hybrid architecture of HyCo2, designed to balance information from richer and more extensive contexts (e.g., top-3 or more documents), might be less optimized for these minimal input scenarios compared to approaches specifically tailored for single-document compression.

**Domain-Specific Generalization.** The current experiments primarily focus on knowledge-intensive question answering tasks, which limits the evaluation scope of $HyCo_2$ to specific domains. Future work should assess the framework's effectiveness across a broader range of applications such as code generation, legal document summarization, or technical report analysis, where context structure and relevance may differ significantly.

**Compression Granularity.** $HyCo_2$ retains approximately 10% of input tokens by default through its classification layer, but this threshold is static and does not dynamically adapt based on content complexity or task-specific requirements. In some cases, particularly with highly nuanced or domain-specific texts, this fixed ratio might discard critical details essential for downstream reasoning.

**Latency in Long-Context Scenarios.** While $HyCo_2$ reduces token usage by an average of 88.8%, the compression process itself introduces additional computational overhead due to the alternating training strategy and dual-path architecture. This can lead to increased latency during inference when dealing with extremely long contexts, potentially offsetting some efficiency gains.

**Scalability with Larger Models.** The current implementation has been tested on LLMs with parameter sizes up to 8B (e.g., LLaMA3.1-8B-Instruct). However, scaling $HyCo_2$ to handle ultra-large models (e.g., those exceeding 13B parameters) or multi-modal architectures could present new challenges in terms of memory footprint, adapter integration, and training convergence.

**Loss of Semantic Nuances.** Despite improved information preservation compared to existing methods like xRAG, soft compression via the hybrid adapter still risks losing subtle semantic nuances embedded in the original text. This limitation becomes more pronounced in contexts requiring deep inferencing, idiomatic understanding, or culturally specific interpretations.

**Dependency on Pretrained Components.** The effectiveness of $HyCo_2$ relies on the quality of underlying pretrained LLMs and their alignment with the hybrid adapter design. Performance may vary significantly when applied to less mature or low-resource language models, particularly for non-English or domain-specific architectures.

## E  PROMPTS USED IN THE EXPERIMENTS

### E.1  TRAINING

To ensure consistency and clarity in pertraining and instruction tuning, we used several prompt templates as shown in Table 10, 11 and 12.

Table 10: Instructions used for Paraphrase Pretraining where [X] and [D] are placeholders for projected feature $\mathcal{V}$ and document D like (Cheng et al., 2024).

- "Background: [X] means the same as [D] "
- "Background: [X] Can you put the above sentences in your own terms? [D] "
- "[X] Please provide a reinterpretation of the preceding background text. [D] "
- "These two expressions are equivalent in essence:(1) [X] (2) [D] "
- "Background: [X] is a paraphrase of what? [D] "
- "[X] Could you give me a different version of the background sentences above? [D] "
- "In other words, background: [X] is just another way of saying: [D] "
- "You're getting across the same point whether you say background: [X] or [D] "
- "[X] After unpacking the ideas in the background information above, we got: [D] "
- "[X] Please offer a restatement of the background sentences I've just read. [D] "
- "Background: [X] , which also means: [D] "
- "Strip away the mystery, and you'll find [X] is simply another rendition of: [D] "
- "The essence of background: [X] is captured again in the following statement: [D] "

Table 11: Instructions used for Completion Pretraining.

- "Using the background [X] , generate a logical and coherent continuation paragraph.",
- "Consider the background [X] . Write the next paragraph that fits the context.",
- "Based on the background [X] , draft a suitable continuation paragraph.",
- "Referencing the background [X] , create a seamless continuation.",
- "Incorporate the background [X] to generate the next segment of the text.",
- "Leverage the background [X] to produce the next logical section.",
- "Using [X] as the background, write the next paragraph.",
- "Generate a follow-up paragraph that incorporates the background [X] .",
- "From the given background [X] , create a continuation paragraph.",
- "Background: [X] ",
- "To provide accurate answers, it's essential to consider the background information presented here. Contextual Background: [X] ",
- "Background: [X] You might find the above background documents helpful.",
- "The following background will help you understand the context for the questions. Please read it carefully before
- responding. Background: [X] ",

Table 12: Instructions used for Instruction Tuning.

- "Refer to the background document and answer the question. Provide only a short answer. Background: [X] Question: {question}"

## E.2    RECONSTRUCT

We use various prompting strategies to reconstruct original context from compressed representations (Dai et al., 2024). As shown in Table 13, these prompts aim to encourage models to rephrase or expand latent semantic representations into natural language text.

Table 13: Prompts used for reconstructing contexts encoded by soft prompt compression method.

- "These two expressions are equivalent in essence: (1) [X] (2)"
- "In other words, background: [X] is just another way of saying:"
- "Background: [X] means the same as"
- "[X] After unpacking the ideas in the background information above, we got:"
- "[X] Please offer a restatement of the background sentences I've just read."

# F    USE OF LLMS

This paper utilized AI assistance for language polishing of the manuscript, including vocabulary correction and spell checking.

