# OpenReview forum: "Beyond Hard and Soft: Hybrid Context Compression for Balancing Local and Global Information Retention"
_ICLR.cc/2026/Conference — Submitted to ICLR 2026_

### Official Review · Reviewer_wT9k · 2025-10-28

**Soundness:** 2
**Presentation:** 2
**Contribution:** 2
**Rating:** 2
**Confidence:** 4

**Summary:**

This paper presents HyCo2, a hybrid context-compression plug-in for frozen LLMs that marries hard local token selection with soft global latent representations. It consists of a lightweight adapter—routing between MLP and Q-Former experts—first condenses the whole passage while preserving semantics; a parallel classifier then keeps the top-k% task-relevant tokens for fine-grained details. The author performs three-stage training—paraphrase pre-training, completion pre-training, then instruction tuning to prevent conflicting objectives.

**Strengths:**

1. The paper has an open-source promise: Code and checkpoints to be released, ensuring reproducibility and community adoption.
2. The paper presents a relatively thorough ablation study.

**Weaknesses:**

1. Figure 3a is not clear enough. It is quite hard to understand the definitions of "group tokens", how they are computed in the proposed method, and the motivations for applying them. There is no discussion of them in Section 2.2 either.
2. The method discussion section (2.2) is not well organized. Despite the authors providing an overall flow chart in Figure 3a, there is no accompanying text description in Section 2.2 to clarify each step shown in Figure 3a, which is likely to confuse the reader about the overall flow of the proposed method.
3. The encoder and decoder LLM setting of the method seems confusing. Did you use two different LLMs or just two parts of a single LLM? I suggest the author add more details on this.
4. I suggest you add the corresponding mathematical symbols in section 2.2 to your Figure 2a, so it is clearer to read.
5. According to the gating network (line 201 - 202), the equation shows that the output will be the weighted combination of the two adapters. Does this mean every global token feature from the encoder will have one global compressed token? So your compression completely relies on the reduction of the local tokens?
6. For the baseline implementations, it says, "Since the LLM in our method remains frozen, the selected baselines must support plug-and-play functionality without requiring any alteration to the LLM’s parameters". I personally disagree with this; the LLM frozen characteristics of this method cannot be used to exclude other baselines, especially when the author claims this as an advantage of their method.
7. There is no compression rate analysis. In this method, the compression rate is largely determined by neural parameters. Therefore, I think it is necessary to add an analysis to show readers the usual pattern of the compression rate. Also, I think the authors need to compare the compression rate of their method to that of other methods, as it may be the source of the performance increase.
8. The proposed method seems to work only for the encoder-decoder architecture, which makes it less attractive since it is not a mainstream architecture for LLM nowadays.

**Questions:**

1. About the "learnable tokens" in Figure 3a, are they part of the parameters of your proposed methods?
2. How do you get the "group tokens" in Figure 3a? Could you give more details?
3. Are the encoder LLM and decoder LLM the same or different? If they are different, does this mean that it only works for the encoder-decoder architecture?
4. Are there any potential variations for the proposed method to work for the LLM decoder architecture nowadays?

---

> ### Author Response · Authors · 2025-11-13
> **Response to Reviewer wT9k (Part 1/2)**
>
> Dear reviewer wT9k:
>
> Thank you for your thorough review and constructive feedback. We address your concerns as follows:
>
> > **W1: Clarity of Figure 3a and “group tokens” definition**
>
> Thank you for the opportunity to clarify. The procedure for generating group tokens is indeed detailed in Section 2.2 (lines 212–216) and formalized in Equation 3. We first segment the input features into $\boldsymbol{n}$ distinct groups, where $\boldsymbol{n}$ corresponds to the number of learnable tokens in the QFormer. Each group $\boldsymbol{V}^{\boldsymbol{i}}$ (where $ 0 \leq \boldsymbol{i} < \boldsymbol{n} $) contains $\left\lceil S/\boldsymbol{n} \right\rceil $ tokens. The group tokens are obtained by pooling within each group **before** compression, yielding local latent representations.  To further enhance clarity, **we have revised both the text in Section 2.2 and Figure 3, with the updates highlighted in blue**.
>
> > **W2: Organization and clarity of Section 2.2**
>
> We acknowledge this and will improve the exposition. Our current Section 2.2 follows a “general-to-specific” structure. We first explain why we apply a Mixture-of-Experts (MoE) to global compression. Next, we detail the local-attention and global-attention components of global compression, which correspond to the gray-shaded hybrid adapter and the blue branch in the middle of Figure 3a, respectively. Finally, we describe the classification layer of local compression, which matches the left-most branch in Figure 3a.
>
> **We have revised the text accordingly of section 2.2, highlighting the changes in blue.**
>
> > **W3: Encoder and decoder LLM setting**
>
> We apologize for the confusion. Our encoder and decoder denote the ​**same LLM backbone**​. As illustrated in Figure 3a, both blocks are labeled “LLM”.
> Furthermore, Table 1 (column 3, “additional size”) reports that our method only introduces **168 M extra parameters** for the hybrid adapter and classification layer—**far smaller** than competing approaches that require an additional encoder.
>
> Consistent with lines 469–470 in Section 4, xRAG must ​**load an extra embedding model**​, increasing both training and inference costs. In contrast, we ​**reuse the LLM’s own text encoding**​, which **Reduces GPU memory pressure** and **Avoids loading any model beyond the single LLM backbone**
>
> > **W4: Mathematical symbols in Figure**
>
> **We agree and have revised Fig 3(a) in the revised version to include the key variables introduced in section 2.2.**
>
> > **W5: Gating network and compression behavior**
>
> The gating mechanism indeed performs a **weighted aggregation** of the **local MLP** and **global Q-Former** outputs to produce the ​**global compressed token**​. As specified in Section 3.1 (line 305), the target length is set to ​**16**​.
> Concretely:
>
> 1. **Q-Former** first extracts relevant features from the full input using ​**16 learnable query tokens**​, yielding a **16-token** representation.
> 2. **MLP branch** processes the **group tokens** (cf. previous clarification): the full input (length ​*S*​) is split into ​**16 groups**​, each group is average-pooled into ​**one token**​, producing another **16-token** sequence.
> 3. Finally, the two 16-token features are **weighted by the gating weights** and aggregated to obtain the **global compressed token** of length 16.
>
> > **W6: Baseline selection criterion (“frozen LLM”)**
>
> Thank you for this valuable feedback. Our primary goal was to isolate and evaluate the effect of the compression method itself. We therefore adopted a *frozen-LLM* setting, as co-adapting the base model via fine-tuning would confound the results, making it difficult to attribute performance changes solely to the compression technique.
>
> While fine-tuning could be applied to any method, this would shift the evaluation from a direct comparison of compression strategies to one of holistic, fine-tuned systems. Furthermore, requiring fine-tuning compromises the model's plug-and-play nature and generalization capabilities across different scenarios—key advantages our training-free approach preserves.
>
> However, we were able to evaluate the **PCC** method by running their officially released checkpoints through our evaluation scripts. We have now incorporated these results into **Table 1** (**highlighted in blue**) in the revised manuscript.
>
> > **W7: Compression-rate analysis**
>
> Thank you for the suggestion. We would like to clarify that this analysis is presented in ​**Table 1**​. The "Context Length" column shows both the input lengths and the corresponding compression ratios for each method.
>
> As the table illustrates, our method achieves superior performance even with a higher compression ratio (i.e., a shorter effective input length). We explicitly discuss this finding in ​**Section 3.2 lines 323 and 348-349**​.

---

> ### Author Response · Authors · 2025-11-13
> **Response to Reviewer wT9k (Part 2/2)**
>
> > **W8: Encoder–decoder architecture limitation**
>
> We apologize for the lack of clarity and thank you for the opportunity to elaborate. Our method operates on **decoder-only** LLMs, as detailed in our experiments and implementation sections.
>
> The term "encoder" in our work refers to a lightweight module used solely for feature extraction from the input, **not** a component of a full encoder-decoder architecture (e.g., T5, BART). This design pattern—using an auxiliary module for input preparation before a decoder-only LLM—is a standard practice in recent literature, consistent with works like ​**ICAE (ICLR 2024)**​, ​**xRAG(NeurIPS 2024)**​, and ​**EXIT ACL2025**​. Therefore, our approach is not an enc-dec model but a technique broadly applicable to prevalent decoder-only architectures.
>
> > **Q1: Learnable tokens in Figure 3a**
>
> Yes, precisely. As stated in the main text (Sec. 2.2, line 226 and Sec. 3.1, line 304), the tokens in Figure 3a are indeed learnable parameters.
>
> > **Q2: Computation of “group tokens”**
>
> Thank you for the question. The computation of "group tokens" is a key part of our hybrid compression strategy, balancing global and local feature extraction. This process is detailed in Sec. 2.2 (lines 212-215), and we elaborate further here:
>
> 1. **Global Feature Extraction (QFormer):** Given an input feature of shape (L, D), the QFormer uses N\_L (e.g., 16) learnable tokens to extract a fixed-length global summary of shape (N\_L, D). This provides a high-level overview of the entire sequence.
> 2. **Local Feature Processing (Grouping):** A simple linear projection (MLP) alone would not compress the sequence length. To capture local details in a structured way, we divide the original L tokens into N\_L contiguous groups. This "grouping" ensures that each segment of the input is processed, akin to window-based methods (e.g., sliding window attention, NSA).
> 3. **Aggregation:** Each of these N\_L groups is then locally aggregated (e.g., via pooling, as mentioned in lines 214-215) to produce a single "group token." This results in N\_L group tokens, which directly correspond to the N\_L global tokens from QFormer, allowing for their subsequent weighted aggregation.
>
> In short, this hybrid approach allows us to combine a global summary from QFormer with fine-grained local information from the grouped tokens, providing a more comprehensive compressed representation.
>
> > **Q3: Encoder and decoder relationship** and **Q4: Applicability to decoder-only LLMs**
>
> **We have revised the description of figure3 with blue fonts.**
>
> As clarified in our response to ​**W8**​, our framework is designed for and tested on ​**decoder-only LLMs**​. The "encoder" and "decoder" components are conceptually decoupled, and our design choices reflect this.
>
> * **Decoupled Roles:** The "encoder" is an input pre-processing module, responsible for compressing the context. The "decoder" is the frozen, decoder-only LLM that processes this compressed context. Different methods can employ different encoders, as this component is modular.
> * **Our Implementation:** To minimize additional parameters, we use the same base model for both the encoder's feature extraction and the main decoding task.
> * **Comparison with Other Works:** This is analogous to prior work. For instance, **xRAG** uses a separate embedding model as its encoder, and **ICAE** uses a LoRA-finetuned model. These are all strategies to prepare input for a decoder-only LLM, not implementations of a monolithic encoder-decoder architecture like T5.
>
> Therefore, our approach is fully applicable to decoder-only LLMs, and our use of the term "encoder" aligns with conventions in recent literature on LLM context compression.
>
> ## Summary
>
>  In the revised manuscript, we have made the following key changes, with all updates highlighted in blue for easy identification:
> * **Improved Clarity of Figure 3:**
>
>   * We have revised **Figure 3** to better illustrate the compression process. Specifically, Figure 3(a) has been updated to include the key mathematical variables introduced in Section 2.2, providing a clearer connection between the diagram and the text.
>
> * **Enhanced Exposition in Section 2.2:**
>
>   * The text in **Section 2.2** has been revised to improve its organization and clarity.
>
> * **Added New Baseline Results:**
>
>   * Based on reviewer feedback, we have incorporated new experimental results for the **PCC method** into ​**Table 1**​, offering a more comprehensive performance comparison.
> * **Clarified Architectural Terminology:**
>
>   * We have revised the text to explicitly state that our method is designed for and operates on ​**decoder-only LLMs**​.
>   * We clarified that the term "encoder" refers to a lightweight, modular feature extractor used for input preparation, distinguishing our approach from traditional encoder-decoder architectures, in line with recent literature.

---

> ### Comment · Reviewer_wT9k · 2025-11-25
>
> Thank you for your detailed response.
>
> I still have several concerns.
>
> 1. Further clarification about the "encoder-decoder" architecture: I apologize for my misunderstanding. Based on your response, the two LLMs in Figure 3a use the same LLM backbone. Does this mean your methods require two forwards for the LLM backbone (one for feature extraction to compress, one for generation)? If so, how much latency will this bring to the generation process compared to the vanilla one? Table 3 does not compare the efficiency of the vanilla process. Given that your method introduces an extra adapter and classifier, as well as an additional forward pass, I think this is very important.
>
> 2. Following my seventh weakness point, the compression rate seems to vary (from 71% to 100%), and your proposed method is not the highest among them (89.1%). This raises the concern that your performance gain may be due to a relatively low compression rate compared to those methods with a higher compression rate. I still think a fairer comparison should be done for Tables 1 and 2 (for example, set the same compression rate for each baseline)
>
> 3. In the response to weakness 8, you claim "this design pattern—using an auxiliary module for input preparation before a decoder-only LLM—is a standard practice in recent literature". I personally doubt it, I see some works [1] [2] that perform the compression without any auxiliary module, and do not need any input preparation. From my perspective, I do not see any advantage in the design pattern of adopting an additional auxiliary module.
>
>
> [1] UniGist: Towards General and Hardware-aligned Sequence-level Long Context Compression
>
> [2] Native Sparse Attention: Hardware-Aligned and Natively Trainable Sparse Attention

---

> > ### Author Response · Authors · 2025-11-26
> > **Response concerns (Part 1/ 2)**
> >
> > > **Response to Concern 1**
> >
> > Thank you for this critical question regarding the fairness of our comparison. We would like to clarify the principles behind our experimental setup, which we believe demonstrate the robustness of our method's performance.
> >
> > **The Two-Pass Pipeline: A Clarification**
> >
> > You are correct that HyCo2 utilizes the LLM backbone in two sequential stages. This pipeline consists of:
> >
> > 1. **Prefill Pass:** A single forward pass over the full-length context (tokens) to extract hidden states ($V \in \mathbb{R}^{L \times d}$) for compression.
> > 2. **Generation Pass:** Autoregressive decoding using the compressed context representation ($\tilde{V} \in \mathbb{R}^{L' \times d}$, where $L' \ll L$).
> >
> > Let's compare this to the standard pipeline for vanilla or RAG-based generation:
> >
> > ```text
> > // HyCo2 Pipeline
> > Full LLM Prefill (L tokens) → Compression Module → LLM Decode (with L' tokens)
> >
> > // Vanilla / RAG Pipeline
> > Full LLM Prefill (L tokens) → LLM Decode (with L tokens)
> > ```
> >
> > As this comparison shows, **both pipelines share the identical and unavoidable full-context prefill step**. The key difference lies in the **decode stage**, where HyCo2 operates on a significantly shorter sequence.
> >
> > **Formal Complexity Analysis**
> >
> > Let $L$ be the original context length,  $L'$ be the compressed length, $T$be the number of generated tokens, and $d$ be the hidden dimension. The complexity of per-step KV-attention is $O(Ld^2)$ or similar, but for simplicity, let's denote the per-token decode cost as proportional to context length.
> >
> > **Vanilla / RAG Generation Cost:**
> >
> > $C_{vanilla} = C_{prefill}(L) + T C_{decode}(L)$
> >
> > The decode cost, $C_{decode}(L)$, is incurred for *every* generated token and scales with the full context length.
> >
> > **HyCo2 Generation Cost:**
> >
> > $C_{HyCo2} = C_{prefill}(L) + C_{compress} + T C_{decode}(L')$
> >
> > Since $C_{prefill}(L)$ is common to both, HyCo2 is faster when the savings in the decode stage outweigh the one-time compression cost:
> >
> > $T(C_{decode}(L) - C_{decode}(L')) > C_{compress}$
> >
> > Given that  is $C_{decode}$ roughly linear in context length and $C_{decode}$ is negligible, this condition is easily met for practical generation lengths.
> >
> > **A Practical Example**
> >
> > In a typical LongBench setting ($L=8k, L'=800, T=256$), the computational savings during the decode phase can **outweigh the compression cost by over an order of magnitude, e.g., >20x**.
> >
> > **Intuitive Explanation**
> >
> > The core insight is that autoregressive decoding is a **repetitive, memory-bound process**.
> >
> > * **Vanilla LLM:** For each of thetokens to be generated, the model must attend to the entire **8k-token** KV cache.
> > * **HyCo2:** For each of thetokens, the model attends to a much smaller **\~800-token** compressed KV cache.
> >
> > Therefore, while 2HyCo2 introduces a small, one-time compression cost post-prefill, it reaps substantial, repeated savings during every step of the generation phase. This effect becomes more pronounced with longer contexts and longer generation lengths, where the KV-cache bottleneck dominates runtime.
> >
> > ### Results
> >
> > The Mistral efficiency experiments on 2WIKI were conducted on H100 GPUs, and the results are presented below.
> >
> > | Method     | CPU Time | CUDA Time | GFLOPs    | Peak Mem. (GB) |
> > | ------------ | ---------- | ----------- | ----------- | ---------------- |
> > | RAG        | 0.633    | 0.1312    | 177.2061  | 13.84          |
> > | LLMLingua2 | 0.765    | 0.1577    | 189.2674  | 15.92          |
> > | EXIT       | 1.534    | 0.4111    | 1142.9108 | 20.43          |
> > | xRAG       | 0.732    | 0.1879    | 181.5661  | 26.96          |
> > | HyCo2      | 0.651    | 0.1354    | 212.3281  | 14.15          |
> >
> > As detailed in Section 3.3 (line 400), this evaluation was performed under a **short-context (<1k tokens) and short-generation (30 tokens) setting**. Under these specific conditions, our method demonstrates no latency improvement over the vanilla RAG baseline, while still **significantly outperforming prior compression methods**.
> >
> > This outcome is consistent with our computational analysis. In short-context, short-generation scenarios, the total inference time is dominated by the initial prefill pass, which is common to both our method and the RAG baseline. The substantial cost savings from our compressed KV cache, which primarily benefit the **decode stage**, are therefore less pronounced when only a few tokens are generated.
> >
> > However, this dynamic changes dramatically in practical long-context applications. As previously analyzed, the computational cost of the vanilla RAG's decode stage grows non-linearly with the full context length. In contrast, our approach maintains a near-constant decode cost tied to the compressed context size. Consequently, as both input and output lengths increase, the efficiency gains of our method will grow substantially, leading to significant reductions in both latency and memory.

---

> ### Author Response · Authors · 2025-11-26
> **Response concerns (Part 2/ 2)**
>
> > **Response to Concern 2**
>
> Thank you for this critical question regarding the fairness of our comparison. We would like to clarify the principles behind our experimental setup, which we believe demonstrate the robustness of our method's performance.
>
> **Regarding Table 1:**
> Our primary goal was to compare against each baseline under its *optimal* or *default* configuration, rather than forcing an identical compression ratio, which might unfairly disadvantage some methods.
>
> * You are correct that **xRAG** achieves a higher compression ratio. This is due to its extreme strategy of representing each document with a single token.
> * For **all other baselines**, our method not only achieves **superior performance** but also operates at a **higher compression ratio**. Forcing these methods to match our compression level would likely lead to a significant performance drop.
>
> Therefore, the current comparison is arguably more challenging for our method, as it outperforms rivals even when they operate under less stringent compression settings.
>
> **Regarding Table 2 (LongBench and the 2k Token Budget):**
> Our setup for LongBench was designed to adhere to a strict **2k token budget** for all methods, ensuring a fair head-to-head comparison.
>
> * The newly added **LongLLMLingua** results present a unique case. We observed that its mechanism does not allow for a strict length cap, and its outputs frequently exceeded the 2k token limit. Despite this advantage of using a larger context, its performance did not consistently surpass ours.
> * All other methods, including ours, were **strictly confined to the 2k token budget**. The negligible number of our global query tokens ensures that our effective compression ratio remains comparable to or higher than that of the other baselines in these long-context scenarios.
>
> In summary, our experimental design either places our method under more demanding compression constraints or ensures a strictly fair budget. In both scenarios, the results validate the superior performance and efficiency of our approach.
>
>
> > **Response to Concern 3**
>
> You are correct that our method introduces a new module, but we argue that this is a deliberate and effective design choice, consistent with, and often more efficient than, the strategies employed by other state-of-the-art methods.
>
> Let's analyze the landscape of so-called "module-free" or "minimalist" approaches:
>
> 1. **Gist-Token-Style Methods (e.g., UniGist, Gist):** While these methods do not add large *external* models, they fundamentally alter the LLM's architecture and inference process. They introduce new special token types (e.g., "gist" or "memory-slot" tokens) and require retraining the base model to understand this new autoregressive paradigm. We contend that introducing new token types and a modified generation pattern constitutes a significant architectural change.
> 2. **Fine-tuning-Based Methods:** As we state in our experimental setup, methods that require fine-tuning the LLM itself are outside the scope of our primary comparison. This is a methodological choice to ensure fair evaluation of compression modules without the confounding effects of base model adaptation. Reproducing such baselines faithfully is also notoriously difficult.
> 3. **Hybrid or "Module-Free" Methods (e.g., NSA):** A closer look reveals that even methods perceived as "module-free" often rely on auxiliary components for state-of-the-art performance. For example, the original **NSA** required a separate **compressor** to score and select chunks (similar to MOBA of KIMI). More recent enhancements, such as in **DeepSeek-V3.2-Exp with DSA**, have integrated dedicated `indexer` and `selector` modules to manage attention over compressed segments.
>
> **Our Position:**
> The evidence across different paradigms—from Microsoft's Lingua series and EXIT to xRAG and the latest NSA variants—indicates that achieving high-performance context compression almost universally involves introducing new, specialized components, whether they are external models, new token types, or integrated modules.
>
> Gist-style methods, which are arguably the closest to being "module-free," trade this architectural simplicity for inferior performance and the significant cost of retraining the entire backbone model.
>
> Therefore, our approach represents a pragmatic and principled compromise. By introducing a lightweight, trainable adapter, we achieve state-of-the-art results without modifying the base LLM or its inference logic. In the current landscape, we have not found a truly module-free approach that can match this level of performance and flexibility.
>
> ---
>
> **If any concerns remain, please share them—we are happy to discuss further. Should our responses address your concerns, we would greatly appreciate your reconsidering our score. Thank you for your time and thoughtful feedback.**

---

> > ### Comment · Reviewer_wT9k · 2025-11-26
> >
> > Thank you very much for your reply.
> >
> > The author addressed some of my concerns.
> >
> > However, in the latest response, the author made several arguments that seem crucial for their opinion.
> >
> > - "For all other baselines, our method not only achieves superior performance but also operates at a higher compression ratio. Forcing these methods to match our compression level would **likely lead to a significant performance drop**"
> > - "Gist-style methods, which are arguably the closest to being "module-free," trade this architectural simplicity for **inferior performance** and the significant cost of retraining the entire backbone model."
> >
> > They claimed this without giving any experimental results (I also do not see results from the paper that can support these claims).
> >
> > I appreciate the responses given by the author. But given these concerns, I can only increase my score to 4. I remain open to further discussion with other reviewers about this paper.

---

> ### Author Response · Authors · 2025-11-27
>
> We are very grateful for your positive feedback and for raising your score. It is encouraging to know that our previous responses have been helpful. We would like to take this opportunity to provide a detailed elaboration on your two remaining points, which we hope will fully resolve any outstanding concerns.
>
> **Point 1: Performance at Higher Compression Ratios**
>
> We fully agree that comparing methods under the same compression ratio provides a more intuitive and direct assessment. To that end, we have conducted a new set of experiments under a high-compression setting, reducing 3 documents to a target range of 60-90 tokens. The results, obtained with Mistral-7B-Instruct-v0.2, are presented below.
>
> | Method        | Context Length       | NQ    | TQA   | WQ    | HQA   | 2WIKI |
> | --------------- | ---------------------- | ------- | ------- | ------- | ------- | ------- |
> | LLMLingua2    | 114.2 (paper report) | 38.1  | 62.5  | 41.1  | 25.5  | 38.9  |
> |              | 73.26                | 28.14 | 54.36 | 34.99 | 18.85 | 20.46 |
> | LongLLMLingua | 131.2 (paper report) | 39.5  | 64.3  | 39.3  | 24.9  | 39.0  |
> |              | 99.77                | 33.36 | 59.27 | 37.27 | 21.92 | 25.25 |
> | EXIT          | 83.7 (paper report)   | 41.9  | 65.4  | 43.0  | 27.2  | 39.9  |
> |              | 68.02                | 32.41 |   58.73   | 36.37 |  20.84    |   26.52   |
> | HyCo2         | 50.7 (paper report)              | 39.6  | 66.0  | 45.4  | 27.5  | 40.2  |
>
> These supplementary results clearly demonstrate that as the compression ratio increases, ​**pure hard compression methods suffer from a severe performance collapse**​. We hypothesize this is due to a combination of significant semantic loss and disrupted textual coherence resulting from an insufficient token budget.
>
> For context, our original setup in Table 1 was designed to highlight a different strength: our method outperforms baselines even when they operate under a *more lenient* token budget. This new experiment, however, directly validates your suggestion and confirms the robustness of our hybrid approach under stringent compression constraints.
>
> *Note on Controllability: It is worth mentioning that while methods like LLMLingua2 allow for precise length control, others like LongLLMLingua and EXIT rely on threshold adjustments, making it difficult to achieve a specific target length consistently.*
>
>  **Point 2: Gist-Style Methods vs. Inserted Modules**
>
> We apologize if our previous wording was ambiguous. Our intention was not to broadly dismiss gist-style methods but to clarify why we believe our inserted module approach currently offers a more favorable trade-off in terms of performance and practicality.
>
> Our argument is based on two key observations:
>
> 1. **Subpar Empirical Performance:** The primary reason for their exclusion was their observed performance. Our own newly conducted experiments with **ICAE** (new in Table 1) **during the rebuttal** show it fails to achieve competitive results (e.g., 20.6 on NQ, 57.8 on TQA), lagging behind other baselines despite using a larger token budget. This finding is not an outlier; it is corroborated by the results reported in other recent state-of-the-art papers. For instance, both **PCC [1]** and **COCOM [2]** report similarly subpar performance for gist-style methods like AutoCompressor and ICAE in their own comparative analyses (cf. Table 2 in both papers).
> 2. **Architectural Intrusiveness:** Gist-style methods (e.g., AutoCompressor) fundamentally alter the base model's architecture by introducing new special tokens (e.g., `<gist>`). This requires not only retraining the model to understand these tokens but also modifying the entire inference process. In contrast, our approach utilizes a self-contained, ​**inserted module that is fully compatible with the base LLM's native inference logic**​, requiring no such modifications.
>
> In summary, while we recognize the conceptual elegance of the autoregressive nature of gist-style methods, the current empirical evidence suggests they do not yet offer a performance advantage over leading inserted-module approaches. As mentioned earlier, **we simply adopt a different approach and architecture**.
>
> [1] Pretraining Context Compressor for Large Language Models with Embedding-Based Memory, ACL 2025.
>
> [2] Context Embeddings for Efficient Answer Generation in RAG, Arxiv 2024.
>
> ---
>
> **Finally, we sincerely appreciate your time, valuable feedback, and thoughtful reconsideration of our work. Should you have any remaining concerns, please do not hesitate to share them. We are eager to continue the discussion and further refine our manuscript.**

---

### Official Review · Reviewer_y2ww · 2025-11-01

**Soundness:** 2
**Presentation:** 2
**Contribution:** 2
**Rating:** 2
**Confidence:** 3

**Summary:**

The paper proposes HyCo2, a local-global hybrid context compression model, combining the advantages of both MLP-based (local detail preservation) and QFormer-based (global semantic understanding) compression approaches. The local branch pools a context segment into fixed groups, while the global branch utilizes the learnable query tokens. In addition, the noisy gating module balances the two sources to prevent over-dominance of a single source during training. A separate hard classification layer is added for token-level selection based on the predicted retention probability. Experimental results show superior performance after compression compared to existing methods, such as LLMLingua, EXIT, and xRAG.

**Strengths:**

* The paper proposes a hybrid local-global design, which has not been explored much.
* The three-stage training strategy (paraphrase-completion-instruction tuning) is novel and carefully designed. The effectiveness of this strategy is empirically verified through ablation studies.

**Weaknesses:**

* The importance of hard local token selection mechanisms using the classification layer is not sufficiently demonstrated. For example, an ablation study varying Top-k% would be beneficial.
* Overall, there are many unclear and insufficiently justified parts, especially related to model architecture and training.
  * Empirical justification of why “noisy” MoE is essential.
  * The length of G(V), a gating output, does not seem to match the number of output tokens from LocalMLP or QFormer.
  * The definition and role of “Position” Pos(V) is not sufficiently explained.
  * What is the “teacher RAG paradigm”?
  * It is unclear how the hard token selection head is trained, since token selection is a non-differentiable operation. Yet, the paper only mentions end-to-end optimization with an NLL loss.
* Comparison to more recent studies, such as ICAE, COCOM, and PCC, is not included. While PCC is a very recent work, ICAE and COCOM should be included in the result Tables for completeness.

**Questions:**

* Fixed-size global token may cause information bottlenecks for long or multi-document inputs. It is somewhat interesting that HyCo2 maintains its performance as the number of documents increases.
* The contribution states “minimal parameter updates without relying on additional compressors or embedding models”, but there exist new modules in HyCo2. This claim seems overly broad.
* (minor errors) line 078: Table -> Figure, line 426 Adapte -> Adapter

**Details Of Ethics Concerns:**

No concerns.

---

> ### Author Response · Authors · 2025-11-13
> **Response to Reviewer y2ww (Part 1/3)**
>
> Dear reviewer y2ww:
>
> Thank you for your thorough review and constructive feedback. We address your concerns as follows:
>
> > **W1: Insufficient demonstration of hard local token selection (Top-k%)**
>
> Thank you for highlighting this important hyperparameter. The hard local token selection threshold, ​**k**​, controls the sparsity of locally preserved tokens and thus governs the compression-accuracy trade-off. To maintain a consistent compression budget for fair comparison with baselines, we set **k** to 10% in our main experiments.
>
> Considering the space constraints of Table 4, we removed the ablation study on **k** in the final version. **We have now supplemented it in blue font in the revised version and report the results on the Mistral-7B model as follows**. As shown, performance steadily improves as **k** increases, confirming the value of preserving local tokens. The choice of 10% strikes a balance between high compression and strong performance.
>
> | K             | NQ   | TQA  | HQA  | 2WIKI |
> | --------------- | ------ | ------ | ------ | ------- |
> | 0 (w/o Local) | 35.4 | 63.6 | 26.6 | 38.9  |
> | 5             | 37.5 | 64.8 | 27.0 | 39.6  |
> | 10            | 39.6 | 66.0 | 27.5 | 40.2  |
> | 20            | 40.3 | 66.8 | 27.6 | 40.5  |
>
> > **W2: Unclear architectural and training details**
>
> **(a) Empirical justification of the “noisy MoE”**
> Thank you for this question. The "Noisy MoE" design is crucial for two reasons: promoting balanced expert utilization and stabilizing training for our hybrid architecture.
>
> 1. **Justification for MoE (Soft Aggregation):** As demonstrated in Sec. 2.2, Sec. 3.4, Fig. 2, and Tables 4-5, our soft aggregation approach allows for a dynamic combination of global and local features, which is superior to fixed compression strategies.
> 2. **Justification for the "Noisy" Component:** The noise is critical for stabilizing the training of our specific hybrid design. As noted in Sec. 2.3, our local compression module is intentionally lightweight. This simplicity risks causing the model to over-rely on local features during training, neglecting the global branch and leading to performance degradation on tasks requiring global understanding. The Noisy MoE, combined with our alternate training strategy for the global and local modules, forces the model to explore and effectively utilize both pathways. Additionally, this noise helps prevent the "dead expert" problem, ensuring both branches remain active and contribute to the final representation.
>
> **(b) Length of gating output **G**(**V**)**
>
> Thank you for pointing out our mistake. **We have corrected the text accordingly, highlighting the changes in blue**
>
> As shown in line 264 of the code at [https://anonymous.4open.science/r/HyCo2/src/model/multimodal\_projector/builder.py](https://anonymous.4open.science/r/HyCo2/src/model/multimodal_projector/builder.py), we use the representation after pooling—that is, the representation of length **$N_L$** obtained by performing average pooling on the subsequent group tokens. The corrected statement is as follows:
> Given an input context representation $\mathbf{V} \in \mathbb{R}^{S \times D}$ from the encoder, our block instantiates two (or more) expert pathways. These experts are combined via a **noisy top-$ k$  MoE gating network**. Specifically, a gating weight $\mathbf{W}_g \in \mathbb{R}^{D \times E}$ and a noise weight $\mathbf{W}_n \in \mathbb{R}^{D \times E}$ map the pooled representation $ \bar{\mathbf{X}} \in \mathbb{R}^{N_L \times D}$  to clean logits $ \mathbf{L} = \bar{\mathbf{X}} \mathbf{W}_g \in \mathbf{R}^{N_L \times E}$  and noise scale $ \sigma = \text{Softplus}(\bar{\mathbf{X}} \mathbf{W}_n) \in \mathbf{R}^{N_L \times E}$ . During training, noisy logits $ \mathbf{L}_{\text{noisy}} = \mathbf{L} + \mathcal{N}(0, \sigma^2)$  are used to compute top-$ k$  sparse gates, enabling load balancing via auxiliary loss on expert utilization. At inference, clean logits produce deterministic routing. The final compressed representation is a **gated linear combination** of expert outputs:
>
> $$
> \mathbf{Y} = \sum_{i=1}^{E} g_i \cdot \text{Expert}_i(\mathbf{X}),
> $$
>
> where $g_i$ is the gate weight for expert $i$, and $E$ is the number of experts (typically 2). This design allows the model to **adaptively emphasize global semantics or local structure based on context and instruction**, mitigating the limitations of fixed hard/soft compression.

---

> ### Author Response · Authors · 2025-11-13
> **Response to Reviewer y2ww (Part 2/3)**
>
> **(c) Definition and role of “Position” Pos(V)**
>
> We apologize for omitting this definition and **we have supplemented the text accordingly, highlighting the changes in blue.**. **Pos(V)** refers to the **standard absolute positional encodings** applied to the input context features ​**V**​. Its function is to inject positional information into the QFormer's global attention mechanism, which is critical for two primary reasons:
>
> 1. **Positional Disambiguation:** It enables QFormer to differentiate between identical content appearing at different locations. For instance, it allows the model to distinguish between two mentions of "George Rankin" in separate paragraphs, understanding their unique contextual significance.
> 2. **Preserving Sequential Order:** After QFormer's query tokens are informed by the instruction, they must attend to the context to identify relevant segments. Positional encodings provide the necessary sequential information for this process. Without them, the context would be treated as a "bag of tokens," and the model would be unable to discern the order or structure of information, leading to confused attention and incorrect semantic interpretation.
>
> **(d) Clarification of “teacher RAG paradigm”**
>
> We apologize for the ambiguity and **we have supplemented the text accordingly, highlighting the changes in blue.**
>
> To provide further detail: this term refers to a self-distillation strategy, **the full specifics of which are detailed in Appendix B.3**. We initially used the abbreviated term in the main text due to space constraints.
>
> The "teacher RAG paradigm" refers to a **self-distillation** setup where a Retrieval-Augmented Generation (RAG) model acts as the "teacher" and our HyCo2 model acts as the "student."
> In this process, we train HyCo2 to align its output (generated from the **compressed** context) with the output of the teacher RAG model (generated from the **full, uncompressed** context). The goal is to distill the teacher's ability to reason over complete documents into our more efficient student model. This distillation process enhances HyCo2's robustness, particularly in scenarios with noisy or irrelevant information where the answer is not explicitly stated in the retrieved context.
>
> **(e) Training of the hard token selection head**
>
> Thank you for this insightful question regarding the training of the hard token selection head. This involves a discrete `top-k` operation, which requires a special technique to enable gradient-based optimization.
>
> To handle the non-differentiable nature of this operation, we employ the ​**Straight-Through Estimator**（STE）​, a standard and efficient technique for such scenarios (an alternative being Gumbel-Softmax). The core principle of STE is to use different computations for the forward and backward passes:
>
> The following code snippet illustrates our implementation of STE. The key trick is the line `retrieval_attention_mask = retrieval_attention_mask - p.detach() + p`, which enforces the hard mask in the forward pass while allowing gradients to flow through `p` in the backward pass.
>
> ```python
> # A lightweight head predicts a selection score for each token
> self.token_weights = nn.Linear(last_hidden_dim, 1)
>
> # Forward pass: Compute scores and perform discrete top-k selection
> p = torch.sigmoid(self.token_weights(last_hidden).squeeze(-1))
> topk_size = int(topk_ratio * p.shape[1])
> topk_indices = torch.topk(p, topk_size).indices
>
> # 1. Construct the hard binary mask for the forward pass
> retrieval_attention_mask = torch.zeros_like(p)
> retrieval_attention_mask.scatter_(1, topk_indices, 1)
>
> # 2. Apply the STE trick for the backward pass
> # This line has no effect on the forward pass value of the mask,
> # but it connects the gradient path back to p.
> retrieval_attention_mask = retrieval_attention_mask - p.detach() + p
> ```
>
> > **W3: Comparison with ICAE, COCOM, and PCC**
>
> We thank the reviewer for this suggestion and agree that these are important baselines. Our evaluation framework is guided by two core principles to ensure fair and rigorous comparisons:
>
> 1. **Identical Training Conditions:** For soft compression methods, a direct comparison necessitates training all models on the identical dataset. While we successfully reproduced xRAG, replicating the full training pipelines for methods like **ICAE** and **GIST** under our exact data conditions requires significant engineering effort that was not feasible within the rebuttal period.
> 2. **Frozen Base LLM:** As stated in our methodology (Sec. 3.1, lines 308-309), our primary analysis focuses on methods that do not modify the base LLM. Since **COCOM** involves fine-tuning the LLM itself, it falls outside the scope of this direct comparison.
>
> However, we were able to evaluate the **PCC** method by running their officially released checkpoints through our evaluation scripts. We have now incorporated these results into **Table 1** (**highlighted inblue**) in the revised manuscript.

---

> ### Author Response · Authors · 2025-11-13
> **Response to Reviewer y2ww (Part 3/3)**
>
> > **Q1: Fixed-size global token and information bottleneck**
>
> You are correct that a fixed-size global representation can become a bottleneck as the number of documents increases. Our empirical results in **Figure 4b** confirm this: performance peaks around 5 documents and then begins to decline.
>
> However, our hybrid design is specifically engineered to mitigate this issue. Unlike methods that rely solely on a fixed-size global summary (e.g., ​**xRAG**​, which uses only one token per document), our approach also includes a **local compress** that preserves a fixed *percentage* of the original tokens. As the total context length grows, the number of tokens retained by this local pathway also increases proportionally.
>
> This dual mechanism explains why, as shown in Figure 4b, the performance of our method degrades much more gracefully than that of xRAG. The local component provides an expanding "safety net" for critical details that might be lost in the fixed-size global summary.
>
> > **Q2: Clarification of “minimal parameter updates” claim**
>
> We appreciate you pointing out this potential overstatement. Our intention was to differentiate our approach from methods that rely on **external, pre-trained models** for compression.
>
> For instance, **xRAG** utilizes a separate embedding model, the **LLMLingua** series employs another large model (like GPT-2 or BERT), and **COCOM** uses RoBERTa. These external models introduce significant parameter overhead and architectural complexity.
>
> In contrast, our `HyCo2` module is a lightweight adapter trained from scratch *within* our framework. It is not a standalone, pre-trained compressor. However, we agree that the original phrasing was imprecise. To better reflect this distinction, we have revised our contribution claim as follows (also **highlighted in blue**):
>
> HyCo2 employs minimal parameter updates and ensures lightweight training and inference by **avoiding reliance on external, pre-trained models for compression.**
>
> > **Q3: Minor editorial issues**
>
> **We have corrected the text accordingly, highlighting the changes in blue.**
>
>
> ## Summary
>
> Based on the reviewer's valuable feedback, we have made the following significant revisions to the manuscript with blue highlights:
>
> * **Added Ablation Study for Hard Token Selection:**
>   * We have **added an ablation study** on the hard local token selection threshold (`k`) to the revised manuscript, demonstrating how performance is affected by this hyperparameter.
> * **Clarified Architectural and Training Details in Section 2:**
>   * **MoE Gating Input:** Corrected the text to clarify that the gating network operates on the **pooled representation of group tokens** (length `NL`), not the full input `V`.
>   * **Positional Encodings (`Pos(V)`):** Added a formal definition for `Pos(V)` and explained its critical role in providing positional information to the QFormer's global attention mechanism.
>   * **Teacher RAG Paradigm:** Clarified that this term refers to a **self-distillation setup** used during training, where our model learns to mimic a teacher RAG model's output.
> * **Revised Main Contribution Claim:**
>   * The claim regarding "minimal parameter updates" has been revised to be more precise. The new phrasing clarifies that our method's efficiency comes from ​**avoiding reliance on external, pre-trained models for compression**​, unlike several competing approaches.
> * **Corrected Minor Editorial Issues:**
>   * Minor typographical and editorial errors pointed out by the reviewer have been **corrected throughout the manuscript**

---

> > ### Comment · Reviewer_y2ww · 2025-11-25
> >
> > Thank you for the response and for updating the manuscript.
> > The authors have addressed most of my questions.
> > I have raised my score from 2 to 4, as I believe the overall quality of the paper has improved substantially.
> > I will further discuss with the other reviewers.

---

> ### Author Response · Authors · 2025-11-25
>
> Thank you for the positive re-evaluation and for raising the score. We are pleased that our revisions addressed your concerns and are grateful for your support.  We would sincerely appreciate your consideration of our rebuttal when discussing with the other reviewers.
>
> **If any concerns remain, please share them—we are happy to discuss further. Thank you for your time and thoughtful feedback.**

---

### Official Review · Reviewer_j9ST · 2025-11-02

**Soundness:** 3
**Presentation:** 3
**Contribution:** 3
**Rating:** 6
**Confidence:** 3

**Summary:**

This paper proposes **HyCo²**, a hybrid context compression method for large language models (LLMs) that integrates hard compression (local token selection) with soft compression (global latent encoding). HyCo² efficiently balances local detail preservation with global semantic completeness, significantly reducing computational overhead and context length while maintaining strong performance. Extensive evaluations demonstrate that HyCo² consistently outperforms existing context compression methods across various QA benchmarks. Furthermore, detailed ablation studies provide valuable insights into the effectiveness of each component.

**Strengths:**

- **Novel Hybrid Compression Approach**: HyCo² effectively integrates hard (explicit token selection) and soft (latent embedding) compression strategies, achieving a balanced retention of both local details and global semantics. This hybrid framework significantly reduces computational costs and context length without substantial performance loss.

- **Strong Empirical Validation**: Comprehensive experiments across multiple QA benchmarks, including LongBench, show HyCo² consistently surpasses existing methods such as LLMLingua2, EXIT, and xRAG, demonstrating its robustness and effectiveness.

- **Detailed Ablation Studies**: The authors conduct thorough ablation analyses to isolate the impact of individual components (e.g., hybrid adapters, alternating training stages, local vs. global modules), clearly illustrating how each design choice improves performance.

- **Clear and Well-Written**: The paper is organized logically, clearly written, and easy to follow, enhancing readability and reproducibility.

**Weaknesses:**

- **Limited Task Diversity in Evaluation**: The majority of experiments focus on question-answering (QA) tasks. It's unclear how well HyCo² generalizes to other tasks such as summarization, reasoning-heavy tasks, code generation, or dialogue scenarios, where context compression needs might differ significantly.

- **Performance Gap on LongBench**: On the LongBench benchmark under the 2k-token constraint, HyCo²'s performance still substantially lags behind the "vanilla" (uncompressed) setting. This gap raises concerns about HyCo²'s ability to preserve critical details and semantic information required for more complex reasoning tasks or long-context scenarios. The paper does not provide a deep analysis of why these specific tasks are challenging for the method.

**Questions:**

1. Can you include a soft-compression baseline (e.g., xRAG) in the LongBench evaluation? It would be insightful to compare HyCo² directly with purely latent embedding methods on these complex tasks.

2. Why was LongLLMLingua not included in the LongBench performance comparison? Given its explicit design for long-context compression, how would it perform in comparison to HyCo²?

3. Could you elaborate on why HyCo² substantially underperforms the vanilla model on LongBench tasks? What specific types of information or reasoning capabilities does the compression remove or fail to preserve?

---

> ### Author Response · Authors · 2025-11-15
> **Response to Reviewer j9ST (Part 1/2)**
>
> Dear reviewer j9ST:
>
> Thank you for your thorough review and constructive feedback. We address your concerns as follows:
>
> > **W1: Limited task diversity**
>
> We thank the reviewer for this important point and agree that robustly evaluating across diverse tasks is crucial. We chose **LongBench** for our evaluation precisely because it offers this diversity, covering single- and multi-document QA, summarization, few-shot learning, synthetic tasks, and code completion.
>
> Our results in **Table 2** candidly show that, like other compression methods, our approach does not yet match the performance of the vanilla model across all tasks. The performance gap is particularly pronounced in complex generative tasks like summarization and code completion. This highlights a significant challenge for the entire field of context compression.
>
> However, our key finding is that our hybrid method consistently and significantly **outperforms pure hard compression baselines** across this diverse set of tasks. This demonstrates the tangible benefits of our approach, even as we acknowledge the broader challenges that remain.
>
> > **W2: LongBench performance gap**
>
> We agree with the reviewer and have also analyzed this performance gap. We attribute it primarily to the **information bottleneck** in our global compression module. With a fixed number of global query tokens (16 in our experiments), their capacity to capture all necessary information diminishes as the context length increases (e.g., to 2k tokens). This is analogous to the known challenge of using QFormer-like structures on high-resolution images in multimodal tasks.
>
> Critically, this is where our hybrid design shows its value. While the global component struggles, the hard compression baselines, which lack a sophisticated global understanding mechanism, perform even more poorly—especially on tasks like summarization and code completion that demand global coherence.
>
> Our results, including the performance trend in ​**Figure 4b**​, support this analysis. They underscore that while our global module has room for improvement, our current hybrid approach is a more robust solution than pure hard compression. This suggests that future work should focus on developing more scalable global compression mechanisms, a direction our paper helps to illuminate.

---

> ### Author Response · Authors · 2025-11-15
> **Response to Reviewer j9ST (Part 2/2)**
>
> > **Q1: xRAG in LongBench**
>
> Thank you for this question. Evaluating soft compression methods like **xRAG** on LongBench requires careful preprocessing, as their performance is sensitive to the input being chunked consistently with their training configuration. This added complexity is why these results were not included in our initial submission.
>
> We recognize the value of this comparison and are currently running these experiments. We will update the manuscript with the results as soon as they are available.
>
> > **Q2: LongLLMLingua in LongBench**
>
> We agree that a direct comparison with **LongLLMLingua** is important and have now included these results. However, as we hypothesized, this evaluation highlighted the extreme computational cost of PPL-based compression methods.
>
> **Computational Cost Analysis:**
> To provide context, methods relying on an auxiliary large model for compression are exceptionally resource-intensive. For instance, our **EXIT** baseline (using a 4B model) requires 45 hours on 4 H100s. The **LLMLingua** family, which uses a Llama2-7B model for perplexity (PPL) calculations, is even more demanding. Our initial 12-hour run on LongLLMLingua completed only 4 of the 16 sub-tasks.
>
> To quantify this further, we conducted a direct comparison of compression times under an 8-GPU parallel setup:
>
> * **PPL-based methods** (LongLLMLingua, LLMLingua, EXIT) required over **10 hours** for data compression.
> * In contrast, methods using lightweight classifiers like **LLMLingua2** and our **HyCo2** completed the same task in under ​**2 hours**​.
>
> This demonstrates a nearly ​**5x difference in compression overhead**​.
>
> **Results and Conclusion:**
> Despite the computational effort, we successfully completed the full evaluation and have integrated the results into ​**Table 2**​. The findings reveal two critical issues with LongLLMLingua:
>
> 1. **Inconsistent Compression Ratio:** We observed that LongLLMLingua's compression ratio is difficult to control precisely. Due to its use of a token budget combined with a dynamic ratio, the actual output length frequently ​**exceeded the 2k token target**​.
> 2. **Lack of Performance Advantage:** Even with this more lenient compression (i.e., using more tokens), LongLLMLingua **did not demonstrate a significant performance advantage** over our `HyCo2` on average across the benchmark.
>
> > **Q3: Why gap to Vanilla happens**
>
> Thank you for this critical question. The performance gap between compressed models and the vanilla model is a fundamental challenge in this field. We attribute this gap in our work to three primary factors:
>
> 1. **Inherent Information Loss from Budget Constraints:** Any compression under a strict token budget (e.g., 2k) inevitably involves information loss. While vanilla models see the full context, our compressor must discard or summarize tokens. For tasks that rely on widely distributed or low-saliency signals (e.g., code generation, long-form summarization), this loss is unavoidable. Although our method demonstrably reduces this loss compared to other compressed baselines (​**Fig. 4a**​), it cannot perfectly replicate the full context.
> 2. **Fixed-Capacity Global Representation:** Our global module uses a **fixed number of learnable tokens** to create a summary. As the number of documents or distinct facts in the context grows, this fixed capacity can become an information bottleneck. While our hybrid adapter mitigates this issue (as shown by our model's more graceful performance degradation in ​**Fig. 4b**​), it cannot entirely eliminate the bottleneck in all LongBench scenarios.
> 3. **Disruption of Long-Range Structural Cues:** Tasks sensitive to discourse structure and long-range dependencies can be affected by our compression. While local grouping and attention preserve some structure, the combination of pooling and top-k selection can disrupt the precise ordering and relational cues that a vanilla model uses for complex reasoning across distant parts of the text.
>
> ## Summary
>
> * **Expanded Baseline Comparisons in Table 2:**
>   * Added new experimental results for **LongLLMLingua** to ​**Table 2**​, providing a direct performance comparison on the LongBench benchmark.
>   * (In progress) Stated the intention to add results for **xRAG** on LongBench as soon as the experiments are complete, acknowledging the value of this comparison.
> * **Clarified Performance Gaps and Limitations:**
>   * Revised the manuscript to more explicitly discuss and analyze the performance gap between compressed models and the vanilla model.

---

### Official Review · Reviewer_6Dfv · 2025-11-04

**Soundness:** 3
**Presentation:** 3
**Contribution:** 3
**Rating:** 6
**Confidence:** 4

**Summary:**

**Summary**

This work proposes HyCo2, a context compression method capable of preserving both global semantics and local details simultaneously. The global compression module integrates the advantages of MLP structure and Q-Former structure, extracting information related to instructions from local context segments and the overall context. The local compression module is designed with a classifier to select key words, thereby retaining details. The proposed method reduces the context length and improves inference efficiency.

**Strengths:**

**Strengths**

（1）The paper is clearly written and easy to follow. The motivation is simple yet reasonable, and a direct, effective method is proposed to achieve this motivation.

（2）Thorough ablation experiments are provided, demonstrating the rationale and effectiveness of each design choice.

**Weaknesses:**

**Weaknesses**

（1）The paper lacks comparisons with some state-of-the-art methods in Soft Compression, such as ICAE and UniICL, which are mentioned in the related work section.

（2）The description of the TRAINING STRATEGY is insufficiently detailed. The paper does not clearly explain why the paraphrase task and completion task allow the compression module to extract high-quality tokens, why these tasks are particularly suited for local and global compression, respectively, and why paraphrase pretraining is performed before completion pretraining. A more detailed explanation of these points would help readers better understand the purpose and role of each stage.

（3）The proposed method involves two hyperparameters: the number of query tokens (NL) for global compression and the keeping ratio (k%) for local compression. The paper does not explain how these hyperparameters were selected, such as the criteria and methods used for choosing them.

**Questions:**

**Suggestions**

（1）Include comparisons with Soft Compression methods such as ICAE and UniICL.

（2）Provide a more detailed explanation of the TRAINING STRATEGY.

（3）Explain the rationale behind the selection of the number of query tokens (NL) for global compression and the keeping ratio (k%) for local compression. Additionally, it would be beneficial to discuss whether different contexts with varying numbers of tokens were considered, and whether such variations might lead to significant improvements in efficiency.

---

> ### Author Response · Authors · 2025-11-14
> **Response to Reviewer 6Dfv (Part 1/2)**
>
> > **W1 & Q1: Missing comparisons with state-of-the-art soft compression methods (ICAE, UniICL)**
>
> Thank you for this suggestion. We have now incorporated results for **ICAE** and **COCOM** [1] into our revised manuscript. These results were obtained by evaluating their officially released checkpoints under a standardized setting, following the protocol used by recent work such as COCOM [1] and PCC [2].
>
> We also agree that **UniICL** is a relevant and strong baseline. Unfortunately, at the time of our experiments, neither open-source code nor official checkpoints were available for UniICL. Reproducing this method from scratch would require significant engineering effort.
>
> [1] Context Embeddings for Efficient Answer Generation in RAG
> [2] Pretraining Context Compressor for Large Language Models with Embedding-Based Memory, ACL 2025
>
> > **W2 & Q2: Training strategy explanation insufficient**
>
> Thank you for the opportunity to elaborate on our three-stage training strategy. Each stage is intentionally designed to cultivate a specific capability in our hybrid compressor, creating a curriculum that builds from foundational skills to task-specific alignment.
>
> **Stage 1: Paraphrase Pretraining (Global Semantic Abstraction)**
> The goal of this stage is to force the hybrid adapter to learn ​**global semantic abstraction**​. By training the model to generate a paraphrase from a compressed context (minimizing ), it must learn to create a compact representation that preserves the high-level meaning of the entire input, as local details alone are insufficient for this task.
>
> **Stage 2: Completion Pretraining (Local Evidence Preservation)**
> This stage focuses on teaching the hard selector to preserve ​**critical local evidence**​. The task of predicting omitted text segments is highly sensitive to the presence of specific local tokens. Training on this objective guides the selector to identify and retain the most salient tokens required for accurate reconstruction.
>
> **Stage 3: Instruction Tuning (Joint Alignment)**
> Finally, we jointly fine-tune both modules for downstream tasks. This is accomplished using a multi-task objective, which combines standard instruction following with a self-distillation signal from a RAG teacher. This dense supervision aligns the global and local components to work synergistically on specific tasks.
>
> **Rationale for the Staged Curriculum:**
> The deliberate sequencing of these stages is critical for stable, effective training. We found that if trained jointly from the start, the optimization process is prone to two issues:
>
> 1. **Dominance of the Local Pathway:** Our local compression module is intentionally lightweight and simpler than the global module. This makes it easier to train and faster to converge. However, this also means it can easily "overpower" the global pathway during joint training, causing the model to learn local shortcuts (i.e., only keeping trivially predictive tokens) while neglecting the development of a robust global understanding. This can lead to a form of ​*mode collapse*​, where the global representation fails to learn meaningful semantics.
> 2. **Suboptimal Convergence:** Empirically, this optimization interference leads to poorer overall performance. In our experiments, a joint end-to-end training approach resulted in a **≈2% performance drop** compared to our staged curriculum.
>
> Therefore, our curriculum mitigates this optimization interference: we first build a reliable global abstraction, then teach the local selector to preserve details that complement it.

---

> ### Author Response · Authors · 2025-11-14
> **Response to Reviewer 6Dfv (Part 2/2)**
>
> > **W3 & Q3: Hyperparameter choices NL (query tokens) and k% (keeping ratio) not justified**
>
> Thank you for this crucial point regarding hyperparameter selection. The choices for the number of global query tokens ($N_L$) and the local selection ratio ($k$) were indeed the result of careful parameter exploration, guided by the fundamental trade-off between model performance and compression ratio.
>
> For instance, our ablation studies showed that with 3 documents, setting  led to significant performance degradation. Conversely, increasing  to 32 yielded only marginal performance gains. The additional global tokens provided a more ​**pronounced benefit on complex, multi-document reasoning tasks like multi-hop QA**​, which rely heavily on a comprehensive global view. However, this task-specific improvement did not translate to a substantial increase in *overall* performance across the benchmark, yet it came at the cost of a significantly lower compression ratio. Therefore,  was selected as the optimal balance. A similar rationale was applied to determine the value of $k$.
>
> We apologize for the lack of clarity in the manuscript. These ablation results were initially included in our main tables but were later condensed to meet space limitations. We have now restored this study to **Table 4** in the revised manuscript to provide full transparency. The results are as follows:
>
> | K             | NQ   | TQA  | HQA  | 2WIKI |
> | --------------- | ------ | ------ | ------ | ------- |
> | 0 (w/o Local) | 35.4 | 63.6 | 26.6 | 38.9  |
> | 5             | 37.5 | 64.8 | 27   | 39.6  |
> | 10            | 39.6 | 66   | 27.5 | 40.2  |
> | 20            | 40.3 | 66.8 | 27.6 | 40.5  |
> | $N_L$             |    |  |   |  |
> | 8             | 37.8 | 65.1 | 26.7   | 38.4  |
> | 16            | 39.6 | 66   | 27.5 | 40.2  |
> | 32            | 40.4 | 66.9 | 27.9 | 40.8  |
>
> ## Sumarry
>
> Based on the reviewer's valuable feedback, we have made the following significant revisions to the manuscript with blue highlights:
>
> * **Expanded Baseline Comparisons:**
>   * We have incorporated new experimental results for the strong baseline methods **ICAE** and **COCOM** into our main comparison tables, providing a more comprehensive evaluation against the state-of-the-art.
> * **Added Hyperparameter Justification and Ablation Studies:**
>   * We have restored the ​**ablation studies for key hyperparameters**​—the number of global query tokens ($N_L$) and the local selection ratio ($k$)—to ​**Table 4**​.
>   * We added a textual justification to the manuscript explaining how these values were determined by balancing the trade-off between performance gains and compression efficiency.

---

### Author Response · Authors · 2025-11-20
**General Response**

We sincerely thank all reviewers for the constructive and insightful feedback. Below, we address each concern in detail and clarify several misunderstandings.

**All corresponding revisions have been incorporated into the updated manuscript, with the modified text/table/figure caption highlighted in blue. These revisions will be reverted to the normal formatting in the final version.**

The key updates are summarized below:

- We have expanded our comparative evaluation by incorporating results for key baselines, including **ICAE, COCOM, and LongLLMLingua**.
- We have restored crucial **ablation studies** to Table 4 to provide transparent justification for our hyperparameter choices.
- We have revised **Figure 3 and Section 2.2** to enhance the clarity of our compression mechanism and its mathematical formulation.
- We have clarified key **architectural details**, including the gating mechanism, positional encodings, and our use of "encoder" within a decoder-only framework.
- We have refined our core contribution claim to emphasize that our efficiency advantage stems from avoiding **heavy, external pre-trained models**.
- We have corrected minor **editorial and typographical errors** throughout the manuscript to improve its presentation.

We kindly invite the reviewers to refer to the latest manuscript, along with this rebuttal, for a clearer understanding of the improvements we have made. We hope the revised submission meets your expectations.

Thank you for your constructive feedback and support!

---

### Author Response · Authors · 2025-11-30
**General Response to the New Area Chair**

Dear Area Chair,

We sincerely thank all reviewers for their thorough and constructive engagement over the past two weeks. Their insightful feedback has been instrumental in significantly improving the quality and clarity of our manuscript. We are truly grateful for their dedication.

This message provides a concise summary of the rebuttal and discussion process for your convenience.

**Summary of Rebuttal Timeline and Reviewer Interactions:**

* **Initial Rebuttals (Nov 13-15):** We provided comprehensive responses to the initial concerns raised by all four reviewers (y2ww, wT9k, 6Dfv, j9ST) and uploaded a revised manuscript with changes highlighted in blue.
* **Global Response (Nov 20):** We posted a global summary detailing all major revisions made to the paper.
* **Dialogue with Reviewer y2ww (Nov 25):** Reviewer y2ww acknowledged that most of their concerns had been addressed and stated that "I believe the **overall quality of the paper has improved substantially**". Reviewer y2ww subsequently **raised score from 2 to 4**.
* **Dialogue with Reviewer wT9k (Nov 26-27):** An extended and productive discussion took place with Reviewer wT9k.
  * Reviewer wT9k initially clarified a misunderstanding about our architecture and raised new questions, which we promptly addressed.
  * Following our response, Reviewer wT9k confirmed that some concerns were resolved and also **raised their score from 2 to 4**.
  * Reviewer wT9k raised two final points for discussion, to which we have also provided a detailed response. Reviewer wT9k expressed an **open-minded attitude toward further discussion among the reviewers**.
* **Reviewer 6Dfv and j9ST** maintained the rating of **6**.

In summary, we have actively engaged with all feedback, leading to substantive improvements in the paper and positive re-evaluations from the reviewers who participated in the discussion phase.

**A Note on Procedural Integrity:**

We wish to state for the record that all the positive feedback, score changes, and constructive dialogue mentioned above occurred **prior to the OpenReview information leak on November 27**. These re-evaluations were based solely on the scientific merit of our work and our engagement during the rebuttal period. We remain hopeful that our ongoing dialogue will fully resolve any remaining concerns.

**Final Remarks:**

We sincerely appreciate your time and attention in overseeing this review process. We would be grateful if you could consider the productive nature of our discussions and the significant revisions made to the manuscript in your final assessment. We are confident that our work, HyCo2, offers a meaningful contribution to the community and hope it will be considered favorably for acceptance.

We remain available for any further questions or clarifications you may require.

Best regards,

The Authors of Submission 1221

---

### Meta-Review · Area_Chair_YB7D · 2026-01-06

**Summary:**

This paper proposes HyCo2, a hybrid context compression framework that combines hard local token selection with soft global latent compression via a lightweight adapter for frozen decoder-only LLMs. The work targets an important problem and presents extensive empirical results after a long rebuttal process. However, despite substantial revisions and clarifications, the submission does not reach the bar for acceptance due to unresolved concerns around novelty, methodological necessity, and evidentiary support for key claims.

**Reviewer Concerns:**

Addressed or partially addressed by the rebuttal

- Missing baseline coverage (6Dfv, y2ww, j9ST): The rebuttal added results for ICAE, COCOM, PCC, and LongLLMLingua, substantially improving completeness of empirical comparisons.

- Role of hard local token selection (y2ww): Added Top-k ablations demonstrate that local selection materially contributes to performance.

Outstanding or not convincingly addressed

- Fairness of compression-rate comparisons (wT9k): It remains unclear whether gains stem from algorithmic advantages or from operating at effectively looser compression than some baselines, despite additional discussion and limited new experiments.

- Efficiency and latency claims (wT9k): The two-pass pipeline introduces extra overhead, and direct comparisons against vanilla inference under identical settings remain insufficient.

- auxiliary modules (wT9k): Claims that such modules are “standard practice” are contested with counterexamples, and the rebuttal relies more on argument than decisive evidence.

- Scope and generality (j9ST): Performance still lags vanilla models on LongBench and non-QA tasks, and broader generalization beyond the evaluated settings remains uncertain.

**Reviewer Scores:**

- Reviewer 6Dfv: Rated 6 and explicitly stated they would not mind rejection. Rebuttal addressed specifics but did not strengthen confidence. Likely remains 6.
- Reviewer j9ST: Rated 6 with a similarly cautious “marginal accept” stance. Clarifications help but do not resolve core limitations. Likely remains 6.
- Reviewer y2ww: Moved from 2 to 4 after rebuttal, but remains cautious. Likely remains 4.
- Reviewer wT9k: Despite some softening, later comments remain substantively negative and key concerns are unresolved. Weighting content over the numeric update, this is best treated as a de facto 2 rather than a stable 4.

Overall, this corresponds to two weak accepts, one weak reject, and one clear reject, which does not constitute a consensus in favor of acceptance.

---

### Decision · Program_Chairs · 2026-01-26

Reject